# Synopsis of Ant Genus *Proceratium* Roger, 1863 from China (Hymenoptera, Formicidae), with Description of Seven New Species [note 1]

**DOI:** 10.3390/insects16101060

**Published:** 2025-10-17

**Authors:** Zhuojian Gu, Chen Zhang, Zhilin Chen

**Affiliations:** 1Key Laboratory of Ecology of Rare and Endangered Species and Environmental Protection (Guangxi Normal University), Ministry of Education, Guilin 541004, China; b1400923315@163.com (Z.G.); eudaemoniazee@163.com (C.Z.); 2Guangxi Key Laboratory of Rare and Endangered Animal Ecology, Guangxi Normal University, Guilin 541004, China

**Keywords:** taxonomy, Proceratiinae, key, new species, China

## Abstract

This study presents a taxonomic synopsis of the ant genus *Proceratium* Roger, 1863 in China, reviewing seven known species (*P. bruelheidei* Staab, Xu & Hita Garcia, 2018; *P. itoi* Eguchi, Yoshimura & Yamane, 2006; *P. japonicum* Santschi, 1937; *P. kepingmai* Staab, Xu & Hita Garcia, 2018; *P. longmenense* Xu, 2006; *P. shohei* Staab, Xu & Hita Garcia, 2018; and *P. zhaoi* Xu, 2000) while removing *P. longigaster* Karavaiev, 1935 from the Chinese fauna. Furthermore, we describe seven new species (*P. crassopetiolum* **sp. nov.**, *P. digitospinum* **sp. nov.**, *P. planodorsum* **sp. nov.**, *P. quandratinodum* **sp. nov.**, *P. rugiceps* **sp. nov.**, *P. shanyii* **sp. nov.**, and *P. soinosubum* **sp. nov.**) and provide an illustrated identification key to Chinese *Proceratium* species. These findings significantly expand our understanding of *Proceratium* diversity in China, resolve taxonomic uncertainties, and establish a foundation for future research on this cryptobiotic genus.

## 1. Introduction

The genus *Proceratium* Roger, 1863 [1] is the most species-rich within the subfamily Proceratiinae, comprising 87 extant and 6 fossil species with a pantropical and temperate distribution [2,3]. Due to their cryptobiotic habits, these ants are rarely encountered in collections [3]. The comprehensive global revision by Baroni Urbani & de Andrade [3] established a critical taxonomic foundation, yet subsequent studies have continued to expand the genus’s known diversity, with numerous new species described [4,5,6,7,8,9]. Despite these advances, Staab et al. [9] highlighted that *Proceratium* taxonomy remains incomplete, with many species still awaiting discovery.

In China, early records were limited to *P. itoi* [10]. Since the 21st century, taxonomic work has accelerated: Xu [5,11] conducted the systematic studies, describing three new species. Liu et al. [12] reported *P. deelemani* Perrault, 1981 [13] from China, later corrected by Staab et al. [9] redescribed it as the new species *P. shohei*. Staab et al. [9] further revised Chinese *Proceratium*, describing three additional species and synonymizing *P. nujiangense* Xu, 2006 with *P. zhaoi* Xu, 2000 based on micro-CT evidence [5]. Their contributions expanded China’s known *Proceratium* diversity from five to eight species while resolving taxonomic ambiguities.

Nevertheless, China’s vast and biodiverse landscapes continue to yield surprises. Here, we describe seven new species: *P. crassopetiolum* sp. nov., *P. digitospinum* sp. nov., *P. planodorsum* sp. nov., *P. quandratinodum* sp. nov., *P. rugiceps* sp. nov., *P. shanyii* sp. nov., and *P. spinosubum* sp. nov. We also review seven previously recorded species and provide an identification key to the Chinese *Proceratium* fauna based on worker caste.

## 2. Materials and Methods

High-resolution color images of the type specimens of three known species (*P. itoi*, *P. japonicum*, *P. longigaster*) were reviewed from AntWeb (https://www.antweb.org). The type specimens of five examined known species (*P. bruelheidei*, *P. kepingmai*, *P. longmenense*, *P. shohei*, *P. zhaoi*) were obtained from the Southwest Forestry University (SWFU). All type specimens of the seven new species are deposited in the entomological collection of Guangxi Normal University, China (GXNU). A total of 28 specimens of the seven new species were collected. Specimens were imaged using a Leica DFC 450 digital imaging system coupled with a stereomicroscope (Leica, Wetzlar, Germany); morphological line drawings were produced using Procreate software (version 5.3.15, Savage Interactive Pty Ltd., Hobart, Tasmania, Australia); image stitching was performed using Adobe Photoshop 2023 software (version 24.0.0, Adobe Systems Incorporated, San Jose, CA, USA). Morphological measurements and indices follow the standardized protocols established by Bolton [14], with minor modifications where necessary. Measurements were recorded in mm and rounded to two decimal places for presentation.

**HL**: Head Length. Maximum measured distance from the anterior-most point of the clypeal margin to the midpoint of a line drawn across the posterior margin of the head.

**HW**: Head Width. Maximum width of head in full-face view (excluding the eyes).

**CI**: Cephalic Index. Calculated as: HW × 100/HL.

**SL**: Scape Length. Excluding the basal radicle.

**SI**: Scape Index. Calculated as: SL × 100/HW.

**ED**: Eye Diameter. Maximum diameter of eye measured in lateral view.

**MSL**: Mesosomal length. Weber’s length measured from the anterior-most point of the pronotal collar to the posterior-most point of the propodeal process.

**PW**: Pronotal Width. Maximum width of mesosoma measured in dorsal view.

**PL**: Petiole Length. Maximum diagonal length of petiole, measured in lateral view, from most anteroventral point of the peduncle, at or below the propodeal lobe, to the most posterodorsal point at the junction with the helical tergite.

**PH**: Petiole Height. Maximum height of petiole, measured in lateral view from the highest point of the node to the ventral outline of the node.

**DPW**: Petiole Width. Maximum width of the petiole in dorsal view.

**LPI**: Petiole length index. Being use to compare the petiolar length-to-width ratio, calculated as: PH × 100/PL. It assesses the length-to-height ratio of the petiole, reflecting its relative height.

**DPI**: Dorsal petiole index. Defined as the ratio of petiole length (PL) to dorsal petiole width (DPW), calculated as: DPW × 100/PL. It quantifies the length-to-width ratio of the petiole, indicating its relative width.

**TL**: Total Length. Maximum length of specimen measured from the tip of the mandibles to the tip of the abdominal segment VII, excluding the sting. Due to the position of the specimen, total length was measured as the sum of head length + mesosomal length, petiole length + gaster length. This article is registered in ZooBank (urn:lsid:zoobank.org:pub:E3249182-A18A-4A78-8AFD-289499DA0B29) in compliance with ICZN requirements for electronic publications. The ZooBank registration was completed prior to final publication.

Below, a drawing showing an entire ant with various morphological features that are mentioned in the text labelled (Figure 1).

## 3. Results

### 3.1. Proceratium Roger, 1863

*Proceratium* Roger, 1863: 171 [1]. Type-species: *Proceratium silaceum*, by monotypy.

*Sysphingta* Roger, 1863: 175 [1]. Type-species: *Sysphingta micrommata*, by monotypy. Junior synonym of *Proceratium*: Mayr, 1886: 437 [15].

**Diagnosis.** (1) Eyes absent or reduced (single convex ommatidium/minute flat ommatidia cluster); (2) antennae with elongated terminal funicular joint (vs. short in *Discothyrea*, *Bradoponera*); (3) mandibles dentate (vs. edentate in *Discothyrea*, *Bradoponera*); (4) maxillary palp with hammer-shaped second joint.

**Distribution.** Afrotropical (11 species), Australasian (10 species), Indo-Australian (20 species), Malagasy (3 species), Nearctic (8 species), Neotropical (20 species), Oriental (8 species), and Palaearctic (16 species) [16].

### 3.2. Synoptic List of China Species of Proceratium

This study documents the following 14 species in China, including 7 new species, as listed below.

*P. bruelheidei* Staab, Xu & Hita Garcia, 2018

*P. crassopetiolum* sp. nov.

*P. digitospinum* sp. nov.

*P. itoi* (Forel, 1918)

*P. japonicum* Santschi, 1937

*P. kepingmai* Staab, Xu & Hita Garcia, 2018

*P. longmenense* Xu, 2006

*P. planodorsum* sp. nov.

*P. recticephalum* sp. nov.

*P. rugiceps* sp. nov.

*P. shanyii* sp. nov.

*P. shohei* Staab, Xu & Hita Garcia, 2018

*P. spinosubum* sp. nov.

*P. zhaoi* Xu, 2000

### 3.3. Key to Chinese Species of Proceratium Based on the Worker Caste

**Notes.** The couplet “10” in the key is adopted from Staab et al. [9]. *Proceratium longigaster* was historically recorded in China, but a re-examination of the specimens underpinning these records revealed they had been misidentified. Although this species is retained in the dichotomous key to facilitate differentiation from morphologically similar Chinese congeners, the taxonomic diagnosis herein formally excludes *P. longigaster* from the Chinese ant fauna. The arrows in Figure 2, Figure 3, Figure 4, Figure 5, Figure 6, Figure 7, Figure 8, Figure 9, Figure 10, Figure 11, Figure 12, Figure 13, Figure 14 and Figure 15 indicate the structures referred to in the associated couplet.
1.Mandibles triangular, masticatory margin with at least 7 teeth (Figure 2A); petiolar node distinctly anteroposteriorly compressed, trapezoidal or rectangular in profile (Figure 2C) …………………………………………………………………………………**2**
Mandibles subtriangular, masticatory margin with 4 large apical teeth (Figure 2B); petiolar node not anteroposteriorly compressed, length equal to or greater than height, peduncle low, cylindrical or bluntly triangular in profile (Figure 2D) ………………………………………………………………………………………………………………………………………………………………………………………………………………………………………………………………………………………………………**7**
Figure 2Head in full-face view and petiole in lateral view. (**A**,**C**) *P. japonicum*; (**B**,**D**) *P. shohei*.
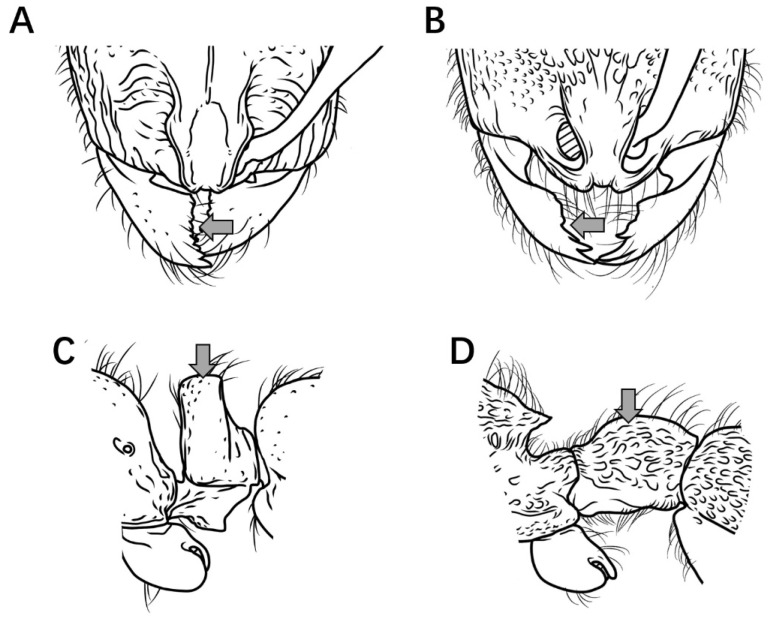

2.Petiolar node rectangular in profile, anterior and posterior margins parallel; junction between propodeal dorsum and declivity rounded, not forming a distinct tooth in profile (Figure 3A) ………………………………………………………..***P. japonicum***
Petiolar node trapezoidal in profile, upper portion distinctly narrower than lower portion; junction between propodeal dorsum and declivity forming a distinct tooth in profile (Figure 3B) ………………………………………………………………………**3**
Figure 3Mesosoma and petiole in lateral view. (**A**) *P. japonicum*; (**B**) *P. longigaster*.
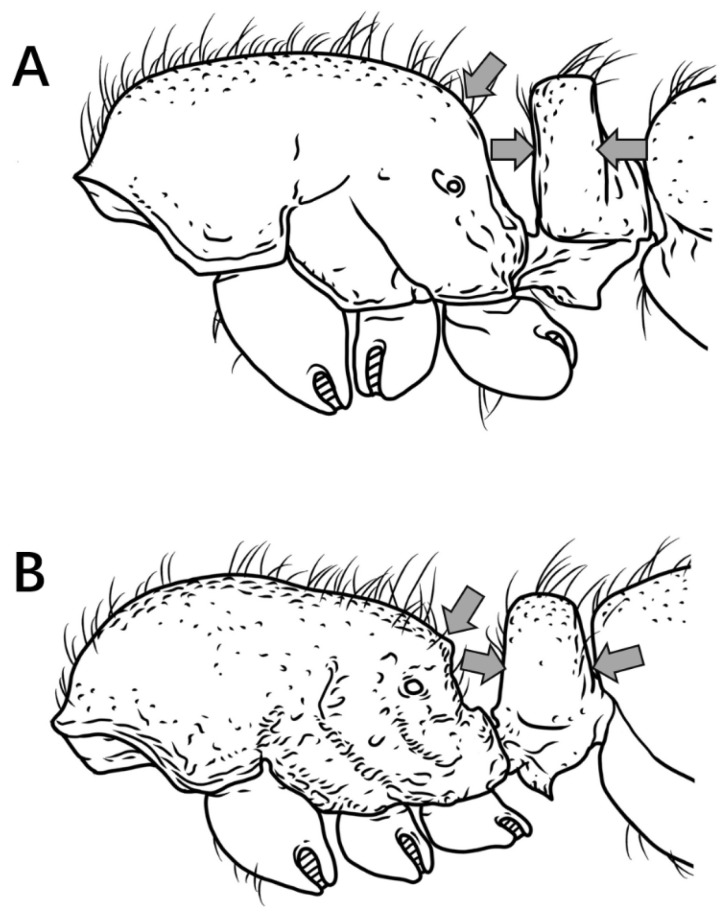

3.Posterior half dorsal surface of head smooth and shining in full-face view (Figure 4A) ………………………………………………………………………………………………………………………………………………………………………………..***P. longigaster***
Posterior half dorsal surface of head sculptured in full-face view (Figure 4B) ……………………………………………………………………………………………………………………………………………………………………………………………………………**4**
Figure 4Head in full-face view. (**A**) *P. longigaster*; (**B**) *P. crassopetiolum* sp. nov.
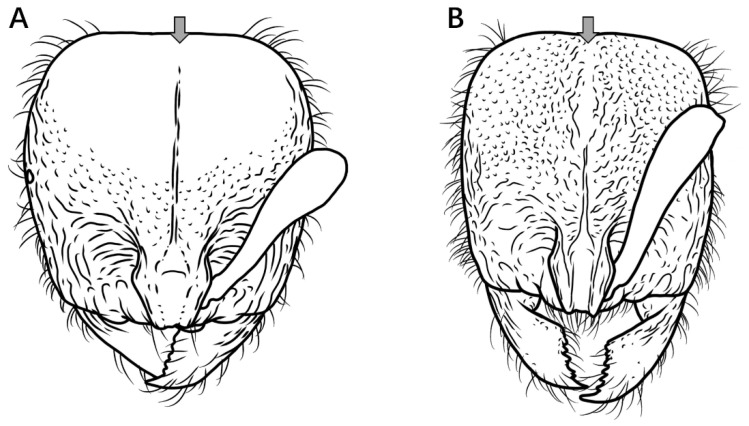

4.Median frontal carina extending posteriorly at most to midlength of head, interrupted by longitudinal or transverse rugae (Figure 5A) ……………………………………………………………………………………………………………………………………...**5**
Median frontal carina extending posteriorly nearly to posterior margin of head (Figure 5B) ………………………………………………………………………………………………………………………………………………………………………………………….**6**
Figure 5Head in full-face view. (**A**) *P. crassopetiolum* sp. nov.; (**B**) *P. shanyii* sp. nov.
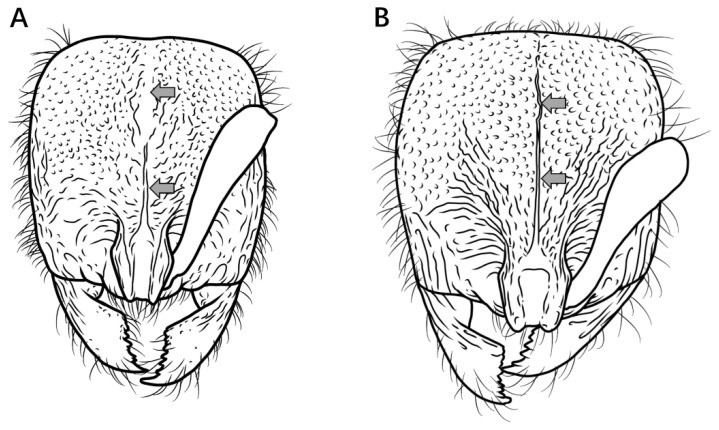

5.Masticatory margin of mandibles with 9–10 teeth; scape relatively long, extending posteriorly to 3/4 the distance from antennal fossa to occipital corner; frontal lobes with a subrectangular shining area, inner margins straight (Figure 6A); petiolar node usually stout in lateral view, height subequal to length, anterior face convex; subpetiolar process spiniform; first gastral tergite with punctures not fused into rugae (Figure 6B) ………………………………………………..***P. crassopetiolum* sp. nov.**
Masticatory margin of mandibles with 7 teeth; scape relatively short, extending posteriorly to 2/3 the distance from antennal fossa to occipital corner; frontal lobes with a non-rectangular shining area, inner margins curved (Figure 6C); petiolar node strongly anteroposteriorly compressed in lateral view, height ≥ 1.5 × length, anterior face steep; subpetiolar process acutely dentiform; first gastral tergite with punctures fused into wavy or ripple-like rugae (Figure 6D) …………………………………………………………………………………………………………………………………………………………………………………………………………………………………………………………………………………***P. rugiceps* sp. nov.**
Figure 6Head in full-face view and body in lateral view. (**A**,**B**) *P. crassopetiolum* sp. nov.; (**C**,**D**) *P. rugiceps* sp. nov.
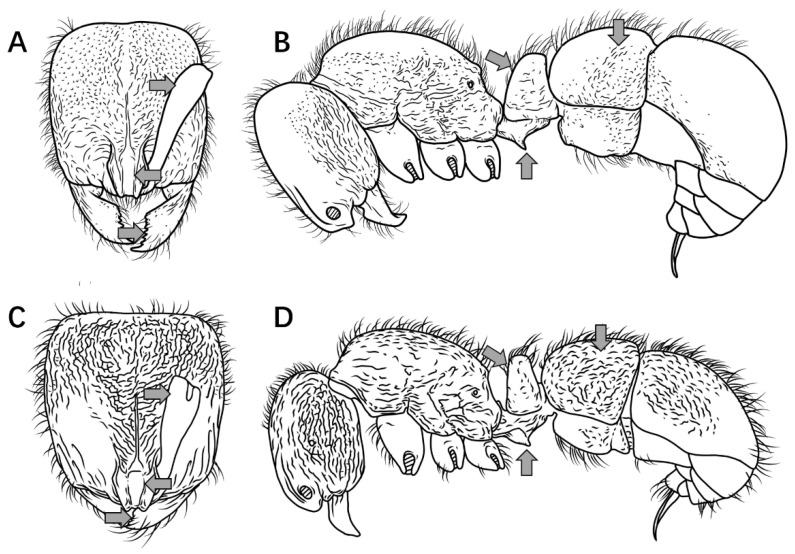

6.Dorsum of head with relatively shallow punctures, lacking reticulation; frontal lobes weakly developed, not conspicuously expanded laterally; rectangular shining area between frontal lobes 1.8 × longer than wide (Figure 7A); mesonotal and metanotal dorsa convex in profile, propodeal declivity steep and anteriorly inclined (Figure 7C); petiolar dorsum narrow, inconspicuous in profile; anterior pronotal margin convex but not hemispherical in dorsal view (Figure 7E) ………………………………………………………………………………………………………………………………………………………………………………………………………………………………………………………………………………….***P. shanyii* sp. nov.**
Dorsum of head with coarse punctate-reticulation; frontal lobes well-developed, conspicuously expanded laterally; rectangular shining area between frontal lobes 2 × longer than wide (Figure 7B); mesonotal and metanotal dorsa flat in profile, propodeal declivity nearly vertical (Figure 7D); petiolar dorsum broad and prominent in profile; anterior pronotal margin strongly rounded, hemispherical in dorsal view (Figure 7F) …………………………………………………………………………………………………………………………………………………………………………………………………………………………………………………………………………...***P. planodorsum* sp. nov.**
Figure 7Head in full-face view, mesosoma and petiole in lateral and dorsal view. (**A**,**C**,**E**) *P. shanyii* sp. nov.; (**B**,**D**,**F**) *P. planodorsum* sp. nov.
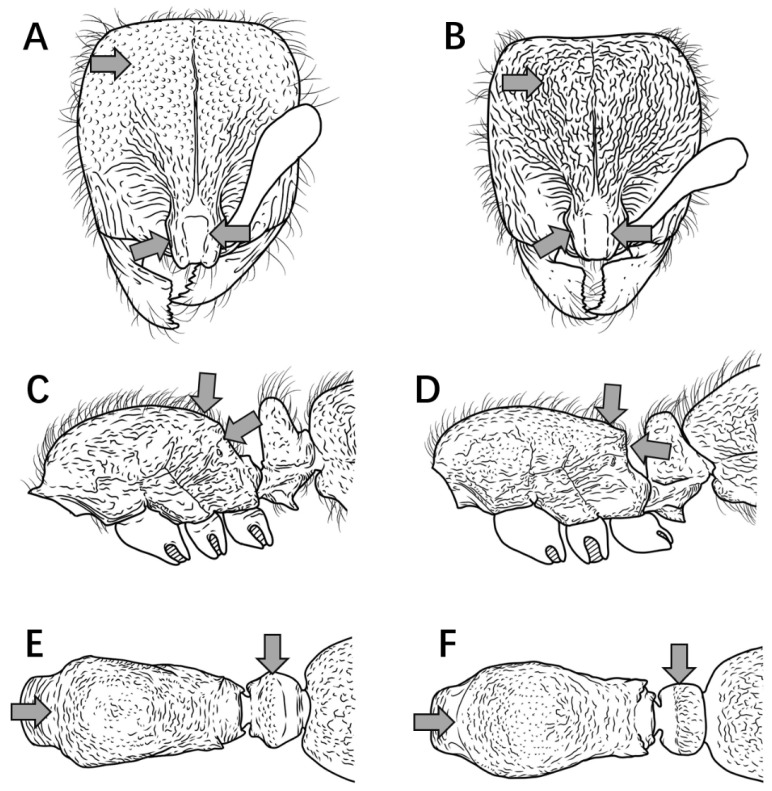

7.Petiole cylindrical, dorsum not nodiform in profile (Figure 8A) …………………………………………………………………………………………………………………………………………………………………………………………………………………………...**8**
Petiole triangular, dorsum nodiform in profile (Figure 8B) …………………………………………………………………………………………………………………………………………………………………………………………………………………………………**9**
Figure 8Petiole in lateral view. (**A**) *P. shohei*; (**B**) *P. longmenense*.
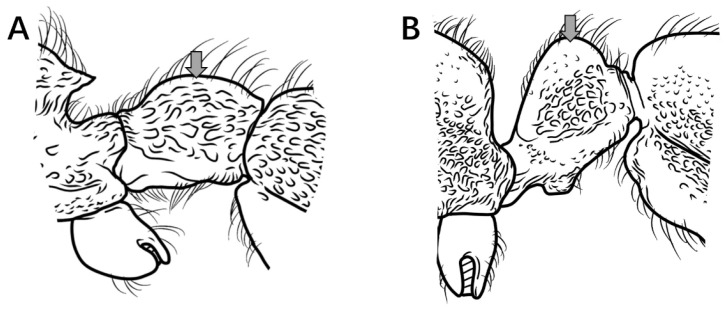

8.Propodeal spines broadly triangular in profile (Figure 9A) ………………………………………………………………………………………………………………………………………………………………………………………………………………………***P. shohei***
Propodeal spines rectangular or digitiform in profile (Figure 9B) ……………………………………………………………………………………………………………………………………………………………………………………………..***P. digitospinum* sp. nov.**
Figure 9Mesosoma in lateral view. (**A**) *P. shohei*; (**B**) *P. digitospinum* sp. nov.
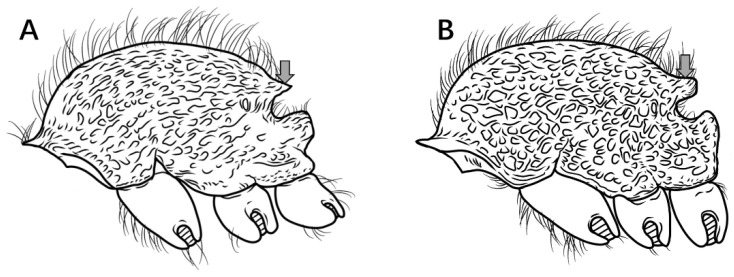

9.Mesosomal dorsum lacking long erect hairs in profile (Figure 10A) ………………………………………………………………………………………………………………………………………………………………………………………………………………***P. zhaoi***
Mesosomal dorsum with abundant long erect hairs in profile (Figure 10B) ……………………………………………………………………………………………………………………………………………………………………………………………………………..**10**
Figure 10Mesosoma in lateral view. (**A**) *P. zhaoi*; (**B**) *P. bruelheidi*.
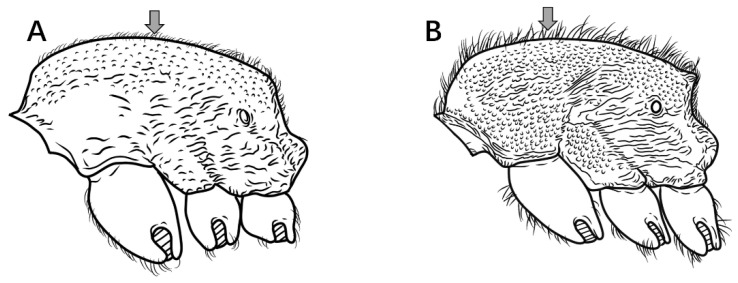

10.Frontal carinae in contact at their anteriormost point, their lateral lamellae relatively narrow, not conspicuously broader above antennal insertions; head relatively narrow (CI = 85) with comparatively long scapes (SI 68) (Figure 11A) …………………………………………………………………………………………………………………………………………………………………………………………………………………………………………………………………………………….***P. longmenense***
Frontal carinae distinctly separated at anteriormost point, their lamellae broad and conspicuously expanded laterally above antennal insertion; head relatively broad (CI ≥ 89) with shorter scapes (SI ≤ 63) (Figure 11B) …………………………………………………………………………………………………………………………………………………………………………………………………………………………………………………………………………………………………….**10**
Figure 11Head in full-face view. (**A**) *P. longmenense*; (**B**) *P. recticephalum* sp. nov.
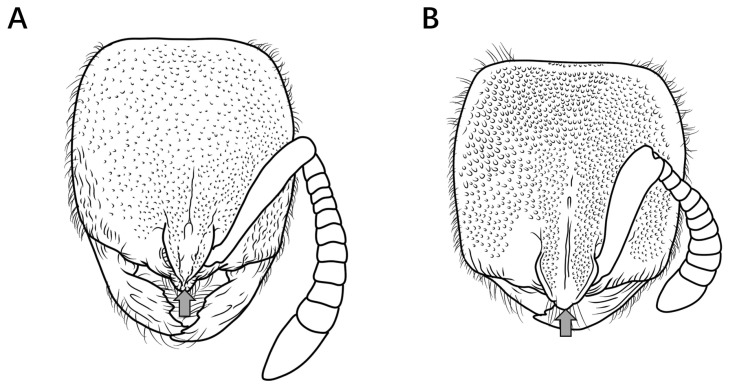

11.Second gastral tergite with very sparse, shallow punctures, appearing smooth and shining in profile (Figure 12A) ………………………………………………………………………………………………………………………………………………………….....**12**
Second gastral tergite entirely covered with coarse punctures or tubercles, appearing dull in profile (Figure 12B) ……………………………………………………………………………………………………………………………………………………………...**13**
Figure 12Gaster in lateral view. (**A**) *P. itoi*; (**B**) *P. bruelheidei*.
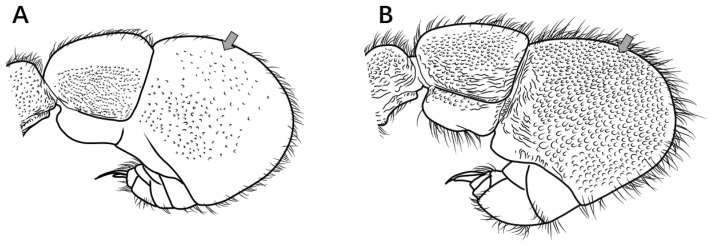

12.No median carina between frontal lobes (Figure 13A); posterolateral mesosomal corners rounded, not dentate in profile; subpetiolar process triangular (Figure 13C) …………………………………………………………………………………………..***P. itoi***
Median carina present between frontal lobes (Figure 13B); posterolateral mesosomal corners dentate in profile; subpetiolar process rectangular, anterior and posterior corners right-angled, ventral margin concave (Figure 13D) ………………………………………………………………………………………………………………………………………………………………………………………………………………………………………………………………………...***P. recticephalum* sp. nov.**
Figure 13Head in full-face view and mesosoma and petiole in lateral view. (**A**,**C**) *P. itoi*; (**B**,**D**) *P. recticephalum* sp. nov.
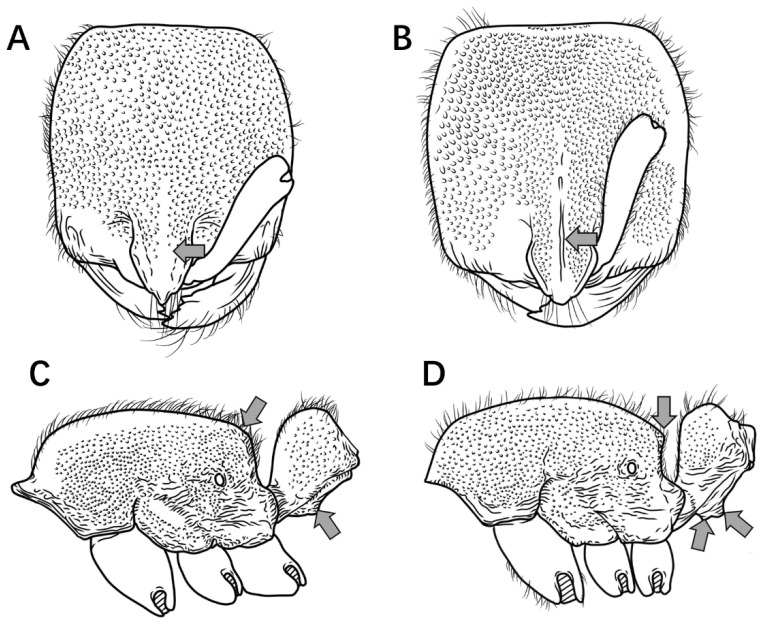

13.In full-face view, posterior head margin convex, head appearing circular (Figure 14A) ………………………………………………………………………………………………………………………………………………………………………………..***P. bruelheidei***
In full-face view, posterior head margin nearly straight, head appearing subrectangular (Figure 14B) ……………………………………………………………………………………………………………………………………………………………………………**11**
Figure 14Head in full-face view. (**A**) *P. bruelheidei*; (**B**) *P. kepingmai*.
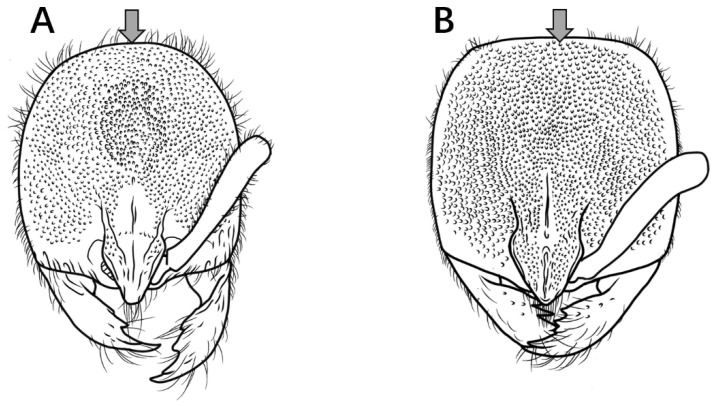

14.In full-face view, lateral carinae of frontal lobe distinctly longer than the distance between the widest points of the frontal lobe (Figure 15A); subpetiolar process rectangular, anteroventral corner indistinct, posteroventral corner bluntly dentate (sometimes appearing broadly triangular); first gastral sternite rectangular, posteroventral margin not concave, anterior portion not lobate (Figure 15C) ………………………………………………………………………………………………….***P. kepingmai***
In full-face view, lateral carinae of frontal lobe distinctly shorter than the distance between the widest points of the frontal lobe (Figure 15B); subpetiolar process spiniform; first gastral sternite not rectangular, anterior 2/3 broader than posterior 1/3, forming a rounded lobe (Figure 15D) ………………………………………………………………………………………………………………………………………………………………………………………………………………………………***P. spinosubum* sp. nov.**
Figure 15PHead in full-face view, petiole and gaster in lateral view. (**A**,**C**) *P. kepingmai*; (**B**,**D**) *P. spinosubum* sp. nov.
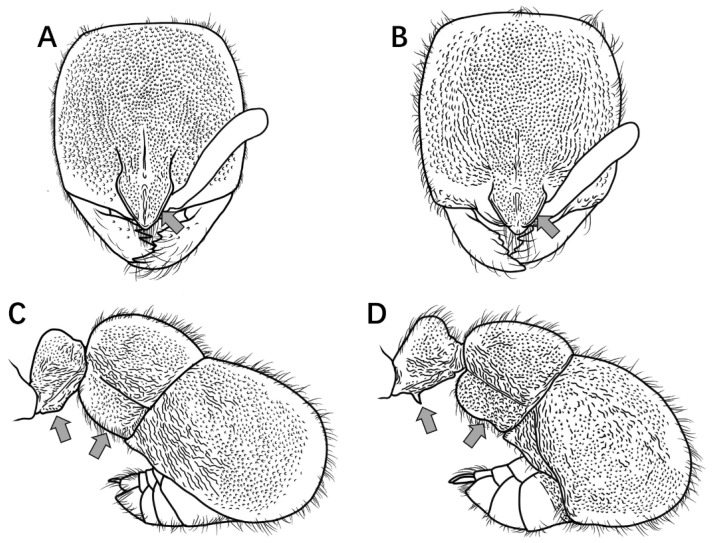


### 3.4. Taxonomic Identifications of the Proceratium Species

#### 3.4.1. *Proceratium bruelheidei* Staab, Xu & Hita Garcia, 2018

Figure 16 and Figure 17

*Proceratium bruelheidei* Staab, Xu & Hita Garcia, in Staab et al., 2018: 148 [9], Figure 4B, Figure 6B, Figure 7B,D, Figure 8 and Figure 9 (w.) CHINA.

**Material examined. *Holotype.*** worker, Xingangshan, Jiangxi, CHINA, 29.1233° N, 117.9069° E, 158 m, 26.iv.2015, leg. Merle Noack, MN290. ***Non-type.*** 1 worker and 1 queen, Tianmushan, Zhejiang, CHINA, 30.3364° N, 119.5031° E, 360 m, 21.vii.2011, leg. Zhilin Chen, GXNU110764.

**Taxonomic notes.** *Proceratium bruelheidei* is most similar to *P. kepingmai*, but can be distinguished by several key features: (1) in full-face view, the head of *P. bruelheidei* has straight lateral margins and a convex posterior margin, while in *P. kepingmai*, the lateral margins are convex, broadest at the level of the eyes, and the posterior margin is almost straight; (2) the propodeal declivity in *P. bruelheidei* is shining and only superficially punctured, compared to the densely punctured, opaque declivity in *P. kepingmai*; (3) the petiolar node in *P. bruelheidei* has a posterior face that is as steep as the anterior face and less than half as long, while in *P. kepingmai*, the posterior face is steeper and about half as long. (4) the apex of the petiolar node in *P. bruelheidei* is slightly broader than long, in contrast to *P. kepingmai*, where it is broader than long.

**Queen measurements.** (n = 1) TL 4.72, HL 0.92, HW 0.86, CI 93, SL 0.56, SI 65, ML 0.46, PW 0.71, MSL 1.35, PL 0.46, PH 0.39, DPW 0.36, LPI 85, DPI 78.

**Description (queen)**. **Head.** In full-face view, head subrectangular, longer than broad (CI 93); lateral margins slightly convex, posterior margin nearly convex; posterolateral corners nearly rounded. Mandibles subtriangular with 3–5 teeth: apical tooth and basal tooth large. Frontal lobes outwardly and upwardly raised, projecting beyond anterior clypeal margin, widest at posterior corners then abruptly narrowing into short frontal carinae; lobes separated by rectangular depression (posterior portion slightly wider than anterior). Antennae 12-segmented; scapes clavate, reaching 0.7× the distance from antennal socket to occipital corner. Eyes large, consisted of more than 80 ommatidium. **Mesosoma.** In lateral view, dorsal outline strongly convex anteriorly, gradually sloping posteriorly; propodeal declivity steep, forming obtuse angle with dorsal margin. In dorsal view, anterior margin rounded; promesonotal margins convex. **Metasoma.** In lateral view, petiole robust (LPI 85), anterior face slightly convex, posterior face nearly vertical; subpetiolar process recurved. In dorsal view, petiole with convex anterior and straight posterior margins; postpetiole wider than long; gaster ovoid. **Sculpture.** Mandibular masticatory margin with scattered shallow piligerous punctures and longitudinally striate. Head with a longitudinal carina extending from frontal triangle to mid-region; central to posterior portion with irregular short wrinkles; lateral surfaces striate. Mesosomal dorsum densely verrucose. Petiole densely verrucose, some linearly arranged. Gastral dorsum with dense, non-confluent verrucae. **Pilosity and pubescence.** Body densely covered with appressed and erect hairs. **Coloration.** Head and mesosoma brown; gaster and legs slightly lighter.

**Distribution:** CHINA (Zhejiang, Jiangxi).

**Habitat.** The species was found in early succulant tree plantation in subtropical mixed forest.

#### 3.4.2. *Proceratium crassopetiolum* sp. nov.


Figure 18


**Material examined. *Holotype.*** worker, Gaozhai village, Guangxi, CHINA, 25.9194° N, 110.4689° E, 1560 m., 12.viii.2021, Zhilin Chen, GXNU209784. ***Paratype.*** 6 workers, with the same data as holotype.

**Worker measurements.** (n = 7) TL 3.10–3.44, HL 0.72–0.76, HW 0.67–0.71, CI 91–96, SL 0.53–0.58, SI 79–84, ML 0.33–0.40, PW 0.51–0.54, MSL 0.84–0.94, PL 0.27–0.31, PH 0.33–0.35, DPW 0.34–0.35, LPI 106–130, DPI 112–126.

**Description (workers)**. **Head.** In full-face view, head subrectangular, longer than broad (CI 91–96); lateral margins slightly convex, posterior margin nearly straight; posterolateral corners obtusely angled. Mandibles subtriangular with 9–10 teeth: apical two teeth large, remaining seven subequal in size, occasionally with basal two teeth fused at base. Frontal lobes outwardly and upwardly raised, projecting beyond anterior clypeal margin, widest at posterior corners then abruptly narrowing into short frontal carinae; lobes separated by rectangular depression (posterior portion slightly wider than anterior). Antennae 12-segmented; scapes clavate, reaching 0.7× the distance from antennal socket to occipital corner. Eyes vestigial, at most a single ommatidium at midline. **Mesosoma.** In lateral view, dorsal outline strongly convex anteriorly, gradually sloping posteriorly; propodeal declivity steep, forming obtuse angle with dorsal margin. In dorsal view, anterior margin rounded; promesonotal margins convex, converging posteriorly at metanotal groove; propodeal margins subparallel with distinct posterolateral denticles. **Metasoma.** In lateral view, petiole robust (LPI 106–130), anterior face slightly convex, posterior face nearly vertical; subpetiolar process recurved; gastral segment I height 0.39× its dorsal length. In dorsal view, petiole with convex anterior and straight posterior margins; postpetiole wider than long; gaster ovoid. **Sculpture.** Mandibular dorsum smooth and shining with scattered shallow piligerous punctures; basal half longitudinally striate. Head with a longitudinal carina extending from frontal triangle to mid-region; central to posterior portion with irregular short wrinkles; lateral surfaces striate. Mesosomal dorsum densely verrucose. Petiole densely verrucose, some linearly arranged. Gastral segment I dorsum with dense, non-confluent verrucae; segment II with shallow sparse punctures, appearing shining. **Pilosity and pubescence.** Body densely covered with appressed and erect hairs. **Coloration.** Head and mesosoma light yellow to yellow; gaster slightly lighter; mandibular masticatory margins dark brown.

**Taxonomic notes.** *Proceratium crassopetiolum* sp. nov. represents a distinct species within the Chinese *Proceratium* fauna, characterized by a unique combination of morphological features that clearly differentiate it from closely related congeners. The species is most readily distinguished by its mandibular dentition (9–10 subequal teeth), trapezoidal petiolar node (height subequal to length with slightly convex anterior face), and discrete punctation on the first gastral tergite that does not form fused rugae. Notably, the median frontal carina extends only to the mid-length of the head, a critical feature that separates it from *P. shanyii*, where the carina reaches nearly the posterior margin. The subpetiolar process is consistently spiniform, contrasting with the dentiform process of *P. rugiceps*, while the scape length (extending to 3/4 head length) provides additional diagnostic value against this latter species. This new species’ sculptured posterior head surface serves as a key distinguishing feature from the superficially similar *P. longigaster* (now excluded from the Chinese fauna), which exhibits a smooth posterior head surface. The rectangular shining area between frontal lobes further differentiates *P. crassopetiolum* from *P. rugiceps* (non-rectangular) and *P. shanyii* (more elongated rectangular). Particularly in cephalic sculpture, petiolar structure, and gastral punctation patterns, provide robust support for recognizing *P. crassopetiolum* as a distinct taxonomic entity. The species’ diagnostic characters remain unambiguous when compared with all currently described congeners, establishing its validity within the genus. The combination of mandibular, petiolar features and non-confluent verrucae of gastral segment I dorsum represents a novel morphological syndrome not observed in other regional *Proceratium* taxa.

**Distribution:** CHINA (Guangxi).

**Habitat.** The species was collected from humid, decaying logs within primary broadleaf forest adjacent to perennial streams. The microhabitat maintains constant humidity year-round, with frequent fog cover persisting through spring, summer, and winter seasons.

**Etymology.** The species name *crassopetiolum* comes from the Latin word *crassus*, meaning “thick”, and the Greek word *petiolum*, meaning “petiole”. The name highlights the species’ most distinguishing feature: a petiole that is unusually thick and stout. This characteristic is the key to its identification and inspired the choice of name.

#### 3.4.3. *Proceratium digitospinum* sp. nov.


Figure 19


**Material examined. *Holotype.*** worker, Huaping village, Guangxi, CHINA, 25.2935° N, 110.1656° E, 160 m, 12.vii.2023, leg. Zhilin Chen, GXNU238761. ***Paratype***. 5 workers, with the same data as holotype.

**Worker measurement.** (n = 6) TL 3.93–3.97, HL 0.97–1.05, HW 0.88–0.96, CI 90–93, SL 0.68–0.76, SI 77–80, ML 0.46–0.52, PW 0.71–0.75, MSL 1.17–1.25, PL 0.49–0.52, PH 0.29–0.32, DPW 0.41–0.46, LPI 59–62, DPI 83–88.

**Description (workers)**. **Head.** In full-face view, head trapezoidal (CI 90–93), lateral margins converging anteriorly, posterior margin convex; posterior corners broadly rounded. Mandibles elongate triangular, masticatory margin with five teeth: apical three large and robust, basal tooth small but distinct, with relatively long gap between subbasal and basal teeth. Clypeus trapezoidal, anterior margin with U-shaped median depression. Scape extending posteriorly to 0.7× head length. Eye reduced to single prominent ocellus, positioned anterior to midpoint of lateral cephalic margin. **Mesosoma.** In lateral view, dorsal outline forming continuous convex arch; posterior propodeum sharply descending, terminating in rectangular projection meeting U-shaped concave propodeal declivity; lateral propodeal lobes well-developed, forming rounded, leaf-like protrusions directed posteriorly and upward. In dorsal view, anterior portion convex with rounded humeral angles; lateral margins converging posteriorly; posterolateral corners forming distinct backward-pointing spines. **Metasoma.** In lateral view, petiole cylindrical (LPI 59–62), lacking distinct nodes; dorsal outline with curved prominence; ventral margin with shallow dentiform projection and anterior concavity. Abdominal segment III slightly shorter than segment IV. In dorsal view, petiolar sides parallel anteriorly; nodal margins convex, posterior margin straight; anterior margin of abdominal segment III straight. **Sculpture.** Mandibular dorsum longitudinally striate. Head, mesosoma, petiole, and abdominal segment III surfaces covered with alveolate pits. **Pilosity and pubescence.** Body with abundant pubescence and erect hairs throughout. **Coloration.** Body uniformly reddish-brown.

**Taxonomic notes.** This new species belongs to the *stictum* clade, comprising 12 species. Only three of these species are known to be distributed in the Oriental Region, namely *P. deelemani*, *P. foveolatum*, and *P. shohei*. However, this new species is easily distinguishable from the known three species in the Oriental region, as it possesses a unique feature—a rectangular extension on the posterior lateral angles of the propodeum in lateral view. We can confirm that this feature is highly stable, as the posterior half of the mesoscutellum abruptly descends forming a distinct concavity, then extends backward into a rectangular prominence. In addition, all paratype specimens obtained from the same nest exhibit this stable characteristic. Furthermore, this new species closely resembles *P. shohei* morphologically, as both species share the same shape of the petiole in lateral view, especially its ventral outline. Moreover, in *P. shohei*, the median tooth on the ventral surface is very indistinct, while *P. deelemani* has a prominent tooth.

**Distribution:** CHINA (Guangxi).

**Habitat.** The species is collected from the soil in the evergreen broad-leaved primitive forests.

**Etymology.** The specific epithet *digitospinum* is a Latin compound adjective derived from *digitus* (finger) + *spina* (spine), referring to the distinctive digitiform (finger-like) propodeal spines that characterize this species. This name highlights the diagnostic morphological feature wherein the propodeal projections are elongated and rounded at the apex (Figure 9B), contrasting sharply with the acutely triangular spines of closely related congeners (e.g., *P. shohei*, Figure 9A).

#### 3.4.4. *Proceratium itoi* (Forel, 1918)


Figure 20


*Sysphincta itoi* Forel, 1918: 717 [10]. Combination in *Proceratium*: Brown, 1958: 247 [17].

**Material examined.** Type specimen unexamined, but high-definition color images of syntype worker were reviewed (https://www.antweb.org/, CASENT0907205, Photographed by Will Ericson).

**Taxonomic notes.** This species differs from other *Proceratium* species in China and neighboring regions by its rounded (vs. dentate) posterodorsal propodeal corners [18]. While most similar to *P. zhaoi*, it can be distinguished by (1) a posteriorly wider head in full-face view; (2) more abundant erect hairs on dorsal body surfaces, and (3) a distinctly triangular (vs. rounded) subpetiolar process. The type locality is confirmed as Japan. Like most congeners exhibiting narrow endemic ranges, the mainland Chinese records require verification. Initial reports from Hunan [19] and Zhejiang [20] lack voucher specimens, and their published illustrations of *P. itoi* depict angular propodeal corners—a non-conforming character. Subsequent records merely cited these unverified reports without original examinations [9,11]. We therefore consider all mainland distribution records questionable until confirmed by examined specimens, recommending the following: re-examination of original collection sites; comparative study with type specimens; molecular verification of putative mainland populations. Additionally, *P. itoi* possesses a highly diagnostic characteristic: in lateral view, the majority of the second abdominal tergite’s surface is exceptionally smooth and shining, allowing it to be easily distinguished from all other species except *P. recticephalum* sp. nov. For morphological differences between *P. itoi* and *P. recticephalum*, refer to the “Taxonomic Notes” following the species description of the latter.

**Distribution:** CHINA (Zhejiang (incredulous), Hunan (incredulous), Taiwan), JAPAN, REPUBLIC of KOREA, VIETNAM (incredulous).

**Habitat.** The species habitat lies in the moist soil and rotting logs of evergreen broad-leaved forests, where the nest and forage within the litter–wood interface.

#### 3.4.5. *Proceratium japonicum* Santschi, 1937


Figure 21


*Proceratium japonicum* Santschi, 1937: 362 [21].

*Proceratium formosicola* Terayama, 1985: 406 [22]. Synonymized by Onoyama, 1991: 695 [23].

**Material examined.** Type specimen unexamined, but high-resolution images of the syntype of *P. japonicum* were reviewed from AntWeb (https://www.antweb.org/, CASENT0915312, photographed by Z. Lieberman).

**Taxonomic notes.** This species is quite distinguishable because its petiolar node in lateral view is rectangular, with the anterior and posterior margins nearly parallel, making it easily distinguishable from species with a narrower apex and wider base.

**Distribution:** CHINA (Taiwan), JAPAN.

**Habitat.** The species was found in evergreen broad-leaved forests.

#### 3.4.6. *Proceratium kepingmai* Staab, Xu & Hita Garcia, 2018


Figure 22


*Proceratium kepingmai* Staab, Xu & Hita Garcia, 2018: 157 [9].

**Material examined. *Holotype***. worker, Xingangshan, Jiangxi, CHINA, 29.1244° N, 117.9611° E, 270 m, 26.iii.2015, leg. M. Staab, MS1836.

**Taxonomic notes.** *Proceratium kepingmai* is most similar to *P. bruelheidei*, but can be distinguished by the following characters: (1) punctate propodeal declivity (vs. smooth in *P. bruelheidei*); (2) more pronounced interfrontal furrow with distinct color contrast against the cephalic background; (3) petiolar node with posterior margin length approximately half of the anterior margin length (vs. subequal in *P. bruelheidei*); (4) additionally, *P. kepingmai* exhibits a more rectangular head shape compared to the subcircular head of *P. bruelheidei*.

**Distribution:** CHINA (Jiangxi, Zhejiang).

**Habitat.** The species was found in secondary subtropical mixed forest.

#### 3.4.7. *Proceratium longigaster* Karavaiev, 1935 (Exclusion from Chinese Fauna)


Figure 23


*Proceratium longigaster* Karavaiev, 1935: 59 [24], Figure 2 (w.) VIETNAM.

*Proceratium longigaster*: Xu 2000: 436 [11], misidentification.

*Proceratium longigaster*: Staab et al. 2018: 172 [9], misidentification.

*Proceratium longigaster*: Liu et al. 2020: 140 [25], misidentification.

**Material examined.** Type specimen unexamined, but high-resolution images of the holotype of *P. longigaster* were reviewed from AntWeb (https://www.antweb.org, CASENT0916806, photographed by Flavia Esteves).

**Taxonomic notes.** The type locality of this species is Vietnam. High-resolution color holotype images of the species can currently be accessed on https://www.antweb.org (CASENT0916806, photographed by Flavia Esteves), as shown in Figure 23. This species is easily distinguishable from all other species in the Oriental region because, in frontal view, the posterior half of its head is smooth and shining, while the dorsum of the promesonotum is highly glossy and lacks sculpturing. Based on currently known species in China, the sculpturing on the body surface of this genus is remarkably stable and can serve as a reliable interspecific diagnostic character. Xu [11] first recorded this species in Yunnan, China, marking its first distribution record in the country. Additionally, Staab et al. [9] reported its occurrence in Zhejiang and Hunan, China. However, upon re-examination of the specimens from Yunnan [11] and the high-resolution images from Zhejiang and Hunan [9], the authors of this study found that their morphological characteristics do not match those of *P. longigaster*. Notably, the posterior half of the head and the dorsum of the promesonotum in the Chinese specimens exhibit distinct sculpturing and are not glossy, contrasting sharply with the holotype of *P. longigaster*. Furthermore, in lateral view, the petiolar node differs significantly: in *P. longigaster*, it appears broader and more robust, whereas the specimens from Yunnan, Zhejiang, and Hunan exhibit a more laterally compressed petiolar node, morphologically closer to *P. shanyii* sp. nov. Staab et al. [9] made significant contributions to the taxonomy of Chinese *Proceratium*, providing high-resolution color images and scanning electron micrographs, as well as clarifying key diagnostic characters among species. However, when identifying the specimens from Zhejiang and Hunan as *P. longigaster*, they did not compare them with the holotype based on the aforementioned morphological differences. The authors of this study argue that the specimens previously identified as *P. longigaster* in China belong to a different species and should not be classified as *P. longigaster*. Therefore, it is recommended that this species be excluded from the Chinese fauna.

**Distribution.** VIETNAM.

**Habitat.** The species was collected from a rotten log in rain forest or secondary subtropical mixed forest.

#### 3.4.8. *Proceratium longmenense* Xu, 2006


Figure 24


*Proceratium longmenense* Xu, 2006: 154 [4].

**Material examined. *Holotype.*** worker, Xishan, Yunnan, CHINA, 24.9501° N, 102.6397° E, 2050 m, 05.v.2001, leg. Zhenghui Xu, SWFU-A00514.

**Taxonomic notes.** This species is most similar to *P. itoi*, but it can be distinguished from the latter by the following combined characteristics: a rectangular head shape (CI = 85), nearly straight posterior margin in frontal view; almost parallel lateral margin, lateral lamellae of frontal carina touching each other at their anteriormost level; scapes relatively long (SI = 68), and a subpetiolar process shaped like a trapezoid; erect hairs are almost absent on the lateral and posterior margins of the head.

**Distribution.** CHINA (Yunnan).

**Habitat.** The species was found in subtropical evergreen broadleaf forest.

#### 3.4.9. *Proceratium planodorsum* sp. nov.


Figure 25


**Material examined. *Holotype.*** worker, Dayaoshan, Guangxi, CHINA, 24.3215° N, 110.1021° E, 230 m, 10.vii.2022, leg. Defu Chen, GXNU238761. ***Paratype***. 1 worker, with the same data as holotype.

**Worker measurement.** (n = 2) TL 3.23–3.24, HL 0.72–0.74, HW 0.72–0.73, CI 97–100, SL 0.52–0.54, SI 72–75, ML 0.38–0.39, PW 0.50–0.51, MSL 0.93–0.94, PL 0.28–0.32, PH 0.33–0.36, DPW 0.30–0.31, LPI 113–118, DPI 97–107.

**Description (workers)**. **Head.** In full-face view, head trapezoidal, narrower anteriorly; lateral margins convex; posterior margin straight; posterolateral corners rounded. Mandibles triangular, masticatory margin with 8–9 teeth: apical two teeth relatively large, teeth 3–7 small and dentiform, teeth 8–9 indistinct. Frontal lobes slightly elevated, anterolateral margins weakly concave; frontal carinae very short, directed posterolaterally; interfrontal space forming rectangular shining area. Antennal scapes 12-segmented, extending posteriorly to approximately 2/3 the distance between antennal fossa and posterior margin of head. Eyes reduced to dark pigmented vestiges, positioned at midlength of lateral cephalic margins. **Mesosoma.** In lateral view, dorsal outline of pronotum convex, anterior slope gradually descending; mesonotum and propodeal dorsum flat; propodeal declivity steep, meeting dorsum at obtuse angulation. In dorsal view, anterior mesosoma semicircular; promesonotal suture faintly impressed; mesosomal lateral margins gradually constricting posteriorly toward propodeum; propodeal lateral margins weakly diverging posteriorly; posterior propodeal margin concave; posterolateral corners acutely angular. **Metasoma.** In lateral view, petiole with anterior margin steep, posterior margin nearly vertical, dorsal margin almost straight. Petiolar node 1.3 times taller than wide; ventral process forming translucent, posteriorly and downwardly directed acute tooth. Abdominal segment III (first gastral) tergite overlapping sternite; anteroventral sternal corner forming rounded, lobe-like projection. In dorsal view, petiole 3 times longer than wide; abdominal segment III narrower anteriorly than posteriorly. **Sculpture.** Mandibles smooth and shining except for piligerous punctures; frontal lobes and interfrontal rectangular area smooth and shining; cephalic midline with distinct carina extending to posterior margin; dorsal cephalic surface densely punctate-reticulate. Lower mesopleura with rugulose-transverse striae, appearing shining; upper mesopleura and mesosomal dorsum densely punctate. Petiolar anterior and lateral faces relatively shining; dorsal petiolar surface densely punctate. Gaster with confluent tuberculate sculpture forming wavy patterns; abdominal segment IV with fine punctures becoming progressively sparser and shallower posteriorly, appearing shining. **Pilosity and pubescence.** Body abundantly covered with pubescence and sub-erect hairs throughout. **Coloration.** Body uniformly yellow, except for blackish-brown masticatory mandibular margins, basal half of antennal scapes, and central posterior cephalic region.

**Taxonomic notes.** *Proceratium planodorsum* sp. nov. is distinguished from other Chinese congeners by its coarse punctate-reticulation covering the dorsal cephalic surface (vs. shallow punctures in *P. shanyii* or smooth posterior head in *P. longigaster*), well-developed frontal lobes conspicuously expanded laterally with a rectangular interfrontal shining area 2× longer than wide (vs. 1.8× in *P. shanyii*), flat mesonotal and metanotal dorsa with nearly vertical propodeal declivity (vs. convex dorsa and anteriorly inclined declivity in *P. shanyii*), strongly rounded hemispherical anterior pronotal margin in dorsal view (vs. convex but not hemispherical in *P. shanyii*), and broad prominent petiolar node in lateral profile (vs. narrow and inconspicuous in *P. shanyii*). The species differs from *P. crassopetiolum* by its longer median frontal carina extending to the posterior head margin (vs. interrupted at midlength) and from *P. japonicum* by its distinct propodeal tooth (vs. rounded junction) and trapezoidal petiolar node (vs. rectangular). Currently known only from China, the species epithet refers to the flattened mesosomal dorsum, a key diagnostic trait. This species belongs to a morphologically complex group where cephalic sculpture, petiolar structure, and mesosomal profile require careful comparison, particularly with historically misidentified specimens of *P. longigaster* excluded from the Chinese fauna.

**Distribution.** CHINA (Guangxi).

**Habitat.** The species was collected from a decaying log (approximately 80 cm in diameter) in a primary broad-leaved forest, with only two worker specimens obtained.

**Etymology.** The specific epithet *planodorsum* is a Latin compound derived from *planus* (flat) and *dorsum* (back), referring to the characteristically flattened mesosomal dorsum that distinguishes this species from its congeners. The name highlights this key diagnostic morphological feature observed in the worker caste.

#### 3.4.10. *Proceratium recticephalum* sp. nov.


Figure 26


**Material examined. *Holotype*.** Worker, Xiangsihu, Guangxi, CHINA, 23.3894° N, 108.3953° E, 200 m, 22.vi.2021, leg. Biao Huang, GXNU218366. ***Paratype*.** 1 worker with the same date as the holotype.

**Worker measurement.** (n = 2) TL 3.93–3.97, HL 0.91–0.93, HW 0.87–0.88, CI 94–96, SL 0.63–0.68, SI 72–78, ML 0.40–0.42, PW 0.69–0.71, MSL 1.16–1.25, PL 0.42–0.44, PH 0.32–0.33, DPW 0.41–0.42, LPI 73–79, DPI 96–98.

**Description (workers)**. **Head.** In full-face view, head rectangular (CI 96), lateral margins nearly parallel, posterior margin straight; posterior corners rounded. Mandibles subtriangular, masticatory margin with four teeth: apical two teeth large and acutely angular, third tooth relatively smaller, basal tooth smallest. Frontal lobes moderately small, anterior margins not converging medially; median carina present between frontal lobes, extending to anterior clypeal margin; frontal carinae short and inconspicuous. Scape extending posteriorly to 3/5 of head length. Eyes reduced to dark pigmented vestiges, positioned at midlength of lateral cephalic margins. **Mesosoma.** In lateral view, mesosomal dorsum flat along middle 3/5, anterior 1/10 and posterior 3/10 sloping downward; propodeal declivity weakly concave, meeting dorsum at obtuse angulation. In dorsal view, anterior pronotal margin semicircularly projecting; lateral margins evenly and slightly converging posteriorly; posterior margin emarginate; posterolateral corners acutely dentate. **Metasoma.** In lateral view, petiole slender, anterior face steep (≥2× longer than posterior face), dorsal profile convex; subpetiolar process rectangular, with distinct right-angled anteroventral and posteroventral corners, ventral margin concave anteriorly. Abdominal segment III (first gastral) relatively small, tergite slightly wider than sternite; anteroventral sternal corner not lobe-like. In dorsal view, petiolar node spindle-shaped. **Sculpture.** Mandibular dorsum with coarse longitudinal striae. Head, mesosoma, petiole, and first gastral segment densely tuberculate. Second gastral tergite and upper half of lateral surfaces with fine shallow punctures, appearing highly shining. **Pilosity and pubescence.** Body densely pubescent with abundant short erect hairs throughout. **Coloration.** Body dark brown; appendages slightly lighter.

**Taxonomic notes.** *Proceratium recticephalum* sp. nov. is distinguished from other Chinese congeners by the following combination of characters: (1) head quadratic in full-face view (CI 96), with nearly parallel lateral margins and straight posterior margin (vs. CI 90 in *P. itoi* and CI ≥ 89 but not quadratic in *P. longmenense*); (2) presence of a distinct median carina between frontal lobes extending to the clypeus (absent in *P. itoi*); (3) rectangular subpetiolar process with right-angled corners and concave ventral margin (vs. triangular in *P. itoi* and spiniform in *P. spinosubum*); (4) dentate posterolateral mesosomal corners (vs. rounded in *P. itoi*); and (5) second gastral tergite with sparse, shallow punctures, appearing highly shining (vs. densely punctate or tuberculate in *P. bruelheidei*). The species is most similar to *P. itoi* but differs conspicuously in cephalic shape (quadrate vs. elongate), mesosomal armature, and petiolar structure. Within the *P. longmenense* species group (characterized by nodiform petiolar nodes and mesosomal erect hairs), *P. recticephalum* is unique in its quadratic head, broad frontal carinae lamellae, and rectangular subpetiolar process. The specific epithet *recticephalum* (Latin: “square-headed”) refers to the distinctive head shape, a diagnostic trait for this taxon. The species is currently known only from China, with ecological preferences likely similar to other *Proceratium* species inhabiting forest litter or decaying wood.

**Distribution:** CHINA (Guangxi).

**Habitat.** The type specimens of this species were fortuitously collected during soil excavation activities in a mixed broadleaf-bamboo forest ecosystem along the Xiangsi hu in Nanning, Guangxi, CHINA. Despite thorough searching of the immediate vicinity, no associated ant colonies were located during the collection event, suggesting either the nest was situated deeper in the substrate or these workers were engaged in extranidal activities at the time of collection. The riparian forest habitat at this location features characteristic subtropical monsoon vegetation with well-developed humus layers, conditions known to support other *Proceratium* species that typically nest in decaying wood or soil organic matter.

**Etymology.** The specific epithet *recticephalum* combines the Latin *recti-* (from *rectangulum*, “rectangular”) and the Greek *-cephalum* (from *kephalē*, “head”), referring to the species’ distinctive rectangular head shape, which corrects the earlier mischaracterization of the head as square (*quadraticephalum*). The name highlights this key diagnostic trait while maintaining nomenclatural consistency with the original description.

#### 3.4.11. *Proceratium rugiceps* sp. nov.


Figure 27


**Material examined. *Holotype*.** Worker, Mengla, Yunan, CHINA, 22.0364° N, 101.8367° E, 920 m, 08.vii.2017, leg. Yunchuan Xiong, GXNU175277. ***Paratype*.** 1 worker with the same data as holotype.

**Worker measurement.** (n = 2) TL 2.42–2.46, HL 0.57–0.59, HW 0.52–0.55, CI 91–93, SL 0.42–0.43, SI 78–81, ML 0.26–0.27, PW 0.40–0.43, MSL 0.68–0.71, PL 0.21–0.23, PH 0.28–0.29, DPW 0.27–0.29, LPI 126–133, DPI 126–129.

**Description (workers)**. **Head.** In full-face view, head slightly trapezoidal, lateral margins convex, posterior margin straight; posterior corners rounded. Mandibles triangular, masticatory margin with seven teeth: apical two teeth relatively large, teeth 3–5 medium-sized, teeth 6–7 basally fused and comparatively smaller. Frontal lobes slightly elevated, anterolateral margins straight; interfrontal space with a shining area narrowing anteriorly and widening posteriorly. Scape extending posteriorly to 3/5 the distance from antennal fossa to occipital margin. Eyes reduced, positioned at midlength of lateral cephalic margins. **Mesosoma.** In lateral view, mesosomal dorsum convex; propodeal declivity steep, meeting dorsum at distinct angulation forming a tooth-like projection. In dorsal view, anterior pronotal margin convex; lateral margins gradually converging posteriorly toward propodeum then slightly diverging; posterior margin emarginate; posterolateral corners acutely dentate. **Metasoma.** In lateral view, petiolar node anteroposteriorly compressed, trapezoidal with both anterior and posterior faces steep; node height approximately 1.2× width. Abdominal segment III (first gastral) tergite overlapping sternite; anteroventral sternal corner forming rounded lobe-like projection. In dorsal view, petiolar node width approximately 4 × length; anterior margin of abdominal segment III nearly straight. **Sculpture.** Mandibular dorsum with longitudinal striae on outer half and sparse punctures on inner half. Triangular area between frontal lobes smooth and shining; remainder of head densely punctate-rugose, with coarsest rugosity along cephalic midline. Mesosomal sides and dorsum with relatively sparse, shallow punctures, somewhat shining. Petiole and first gastral tergite with rugae formed by confluent tubercles. Second gastral tergite smooth and shining. **Pilosity and pubescence.** Body densely covered with decumbent pubescence and abundant suberect hairs throughout. **Coloration.** Body yellowish-brown, head slightly darker in coloration.

**Taxonomic notes.** *Proceratium rugiceps* sp. nov. is distinguished from other Chinese congeners by the following combination of characters: (1) mandibles with seven teeth (vs. 9–10 in *P. crassopetiolum* and 3 large apical teeth in *P. shohei* group); (2) petiolar node strongly anteroposteriorly compressed (height ≥ 1.5 × length) with steep anterior face (vs. height subequal to length in *P. crassopetiolum*); (3) first gastral tergite with punctures fused into distinctive wavy or ripple-like rugae (vs. discrete punctures in *P. crassopetiolum*); (4) frontal lobes with non-rectangular shining area and curved inner margins (vs. subrectangular area with straight margins in *P. crassopetiolum*). The species is most similar to *P. crassopetiolum* but differs conspicuously in dentition (7 vs. 9–10 teeth), petiolar proportions, and gastral sculpture. Within the *P. japonicum* species group (characterized by ≥7 mandibular teeth and compressed petiolar nodes), *P. rugiceps* is unique in its combination of reduced dentition and pronounced gastral rugosity.

**Distribution:** CHINA (Yunnan).

**Habitat.** The species was collected from the decaying wood in subtropical evergreen broad-leaved forest.

**Etymology.** The specific epithet *rugiceps* is derived from the Latin words *ruga* (wrinkle) and *ceps* (head), referring to the distinctive wrinkled cephalic sculpture pattern that characterizes this species. The name highlights the diagnostic rugose-punctate sculpture on the dorsal surface of the head.

#### 3.4.12. *Proceratium shanyii* sp. nov.


Figure 28


**Material examined. *Holotype.*** Worker, Huaping village, Guangxi, CHINA, 25.2930° N, 110.1659° E, 160 m, 28.vii.2020, leg. Zhilin Chen, GXNU20001224. ***Paratype***. 3 workers with the same data as holotype.

**Worker measurement.** (n = 4) TL 2.80–3.13, HL 0.66–0.73, HW 0.60–0.67, CI 91–97, SL 0.47–0.49, SI 73–78, ML 0.29–0.33, PW 0.42–0.48, MSL 0.81–0.87, PL 0.25–0.32, PH 0.28–0.33, DPW 0.28–0.32, LPI 103–112, DPI 100–112.

**Description (workers)**. **Head.** In full-face view, head narrower anteriorly than posteriorly, lateral margins convex, posterior margin straight. Mandibles triangular, masticatory margin with eight teeth gradually decreasing in size from apical to basal. Frontal lobes slightly elevated, anterolateral margins nearly straight; interfrontal space with rectangular shining area. Antennae 12-segmented; scape long, extending posteriorly to approximately 4/5 the distance from antennal fossa to occipital corner. Eyes reduced to dark pigmented vestiges. **Mesosoma.** In lateral view, mesosomal dorsum convex; propodeal declivity descending posteriorly, steep, meeting dorsum at distinct angulation forming a tooth-like projection. In dorsal view, anterior mesosoma convex, posterior portion concave; lateral margins slightly converging posteriorly from mesonotum. **Metasoma.** In lateral view, petiolar node strongly anteroposteriorly compressed (height 1.5 × length), anterior face steep, posterior face vertical; anteroventral sternal corner of first gastral segment forming rounded lobe-like projection. In dorsal view, petiolar node width approximately 4 × length; anterior margin of first gastral segment nearly straight. **Sculpture.** Mandibular dorsum with basal longitudinal striae, remaining surface smooth and shining except for sparse piligerous punctures; arcuate rugulae surrounding antennal fossae and lateral frontal carinae; remainder of head with shallow punctures; cephalic midline with distinct carina extending from frontal triangle to posterior cephalic margin. Mesosomal sides with fine longitudinal-transverse rugulae, appearing shining; rugulae becoming slightly coarser on propodeal sides. Mesosomal dorsum with linearly arranged tubercles anteriorly, transverse rugulae posteriorly. Petiole surface tuberculate; first gastral tergite with irregular or wavy rugae, becoming more distinctly tuberculate posteriorly. Second gastral tergite with very shallow punctures, appearing highly shining. **Pilosity and pubescence.** Body densely covered with decumbent long pubescence and abundant long erect hairs throughout. **Coloration.** Body yellow to yellowish-brown; front slightly darker.

**Taxonomic notes.** *Proceratium shanyii* sp. nov. is distinguished from other Chinese congeners by the following combination of characters: (1) head with relatively shallow punctures lacking reticulation (vs. coarse punctate-reticulation in *P. planodorsum*); (2) weakly developed frontal lobes not conspicuously expanded laterally (vs. well-developed and expanded in *P. planodorsum*); (3) rectangular interfrontal shining area 1.8× longer than wide (vs. 2× in *P. planodorsum*); (4) convex mesonotal and metanotal dorsa in profile with anteriorly inclined propodeal declivity (vs. flat dorsa and vertical declivity in *P. planodorsum*); (5) narrow, inconspicuous petiolar dorsum in profile (vs. broad and prominent in *P. planodorsum*). The species is most similar to *P. crassopetiolum* but differs in its median frontal carina extending nearly to the posterior head margin (vs. interrupted at midlength in *P. crassopetiolum*) and convex dorsal outline of mesosoma (vs. convex anterior face only in *P. crassopetiolum*).

**Distribution:** CHINA (Guangxi).

**Habitat.** The species was collected from decaying wood in tropical rainforest.

**Etymology.** The specific epithet *shanyii* is a Latinized genitive form honoring the late Professor Zhou Shanyi (周善义), the esteemed Chinese myrmecologist who made significant contributions to the study of Formicidae in China.

#### 3.4.13. *Proceratium shohei* Staab, Xu & Hita Garcia, 2018


Figure 29


*Proceratium shohei* Staab et al., 2018: 145 [9].

*Proceratium deelemani*: Liu et al., 2015b: 56 [12], misidentification.

**Material examined.** High-resolution images examined from AntWeb (CASENT0717686). ***Syntype.*** worker, Xishungbanna, Yunnan, CHINA, 21.964° N, 101.202° E, 820 m., 13.vi.2013 leg. B. Guénard, B. Blanchard & C. Liu, CASENT0717686

**Taxonomic notes.** *Proceratium shohei* represents a distinctive member of the Oriental *P. stictum* clade, characterized by the following: (1) broad triangular propodeal spines projecting less than half the propodeal lobe length (vs. ≥ half in *P. deelemani*); (2) inconspicuous subpetiolar process lacking median projection (vs. distinct tooth in *P. deelemani* and *P. stictum*); (3) straight ventral LS3 outline (vs. concave in *P. deelemani*); (4) foveolate sculpture restricted to head, mesosoma, petiole and LT3 (vs. coarsely granulate-foveate in *P. stictum* or deep regular foveae on LT4 in *P. foveolatum*). The species is most similar to *P. deelemani* but differs diagnostically in scape length (SI = 72 vs. 58–68), frontal carinae convexity (slightly convex vs. concave), and propodeal tooth proportions. Unique within the clade for its combination of broad head (CI ≥ 89), elongated scapes (SI = 72), and reduced LS4 preventing IGR measurement. The holotype’s foveolate-punctate sculpture pattern and broadly rounded abdominal segment IV further distinguish it from congeners. Current records suggest this is a rare species endemic to the Oriental region, with ecological preferences likely aligned with other clade members inhabiting forest litter strata. Further collections are needed to assess intraspecific variation and precise distributional range.

**Distribution.** CHINA (Yunnan).

**Habitat.** The species was collected from deadwood in subtropical evergreen broad-leaved forest.

#### 3.4.14. *Proceratium spinosubum* sp. nov.


Figure 30


**Material examined. *Holotype.*** Worker, Tongguling, Hainan, CHINA, 19.6712° N, 111.0171° E, 277 m, 10.iv.2024, leg. Zhilin Chen, GXNU240025. ***Paratype***. 4 workers with the same data as holotype.

**Worker measurement.** (n = 5) TL 3.23–3.24, HL 0.68–0.69, HW 0.62–0.63, CI 91–92, SL 0.42–0.44, SI 68–70, ML 0.31–0.33, PW 0.44–0.46, MSL 0.87–0.89, PL 0.31–0.32, PH 0.25–0.26, DPW 0.27–0.28, LPI 80–82, DPI 87–88.

**Description (workers)**. **Head.** In full-face view, head subcircular, lateral margins convex, posterior margin straight; posterior corners narrowly rounded. Mandibles subtriangular, masticatory margin with four teeth gradually decreasing in size from apical to basal, basal tooth tuberculate. Frontal lobes relatively short, anterolateral carinae not meeting anteriorly, length of lateral carinae shorter than distance between widest points of frontal lobes; frontal carinae inconspicuous. Scape extending posteriorly to 7/10 the distance from antennal fossa to posterior cephalic margin. Eyes reduced to dark pigmented vestiges, positioned at midlength of lateral cephalic margins. **Mesosoma.** In lateral view, mesosomal dorsum evenly convex, abruptly descending posteriorly to meet nearly vertical propodeal declivity at obtuse angle. In dorsal view, anterior mesosoma rounded-convex; lateral margins gradually converging posteriorly; posterior margin emarginate; posterolateral corners acutely dentate. **Metasoma.** In lateral view, petiole relatively slender, both anterior and posterior faces steep (anterior face ≥ 2× longer than posterior face), dorsal profile convex; subpetiolar process spiniform, directed ventrally (slightly stouter in paratype). Anterior half of first gastral sternite forming rounded lobe-like projection, posterior half with concave ventral margin. In dorsal view, petiolar node elliptical. **Sculpture.** Mandibular dorsum with coarse longitudinal striae; head, mesosoma, petiole and gaster densely tuberculate except for smooth and shining second gastral tergite. **Pilosity and pubescence.** Body densely pubescent with abundant short erect hairs throughout. **Coloration.** Body yellowish to yellowish-brown.

**Taxonomic notes.** *P. spinosubum* sp. nov. is distinguished from other Chinese congeners by the following combination of characters: (1) lateral carinae of frontal lobe distinctly shorter than the distance between the widest points of frontal lobe (vs. distinctly longer in *P. kepingmai*); (2) spiniform subpetiolar process (vs. rectangular in *P. kepingmai*); (3) first gastral sternite with anterior 2/3 broader than posterior 1/3, forming a rounded lobe (vs. rectangular with uniform width in *P. kepingmai*). The species belongs to the *P. itoi* clade, within this clade, *P. spinosubum* is most similar to *P. recticephalum* but differs in: (1) frontal lobe carinae proportions; (2) subpetiolar process morphology (spiniform vs. rectangular); and (3) gastral sternite shape. The spiniform subpetiolar process represents a unique autapomorphy among Chinese *Proceratium* species.

**Distribution:** CHINA (Hainan).


**Habitat. The species was collected from decaying wood in tropical rainforest.**


**Etymology.** The specific epithet *spinosubum* is a Latin compound derived from *spina* (spine) and *subum* (beneath), referring to the diagnostic spiniform subpetiolar process that distinguishes this species from its congeners.

#### 3.4.15. *Proceratium zhaoi* Xu, 2000


Figure 31


*Proceratium zhaoi* Xu, 2000: 435 [11].

*Proceratium nujiangense* Xu, 2006: 153, figs. 8–13 (w.q.) CHINA (Yunnan) [5]. Synonymized by Staab, et al., 2018: 164 [9]. 

**Material examined. *Holotype*.** Workers, Mengla, Yunnan, CHINA, 22.9167° N, 101.4167° E, 1280 m, 10.ix.1997, leg. Zhenghui Xu, A97–2338.

**Taxonomic notes.** In the Oriental region, the genus *Proceratium* currently comprises a total of 26 species. Among these, only two species, namely *P. zhaoi* and *P. williamsi*, lack erect hairs on their dorsal surfaces. The measurements provided by Staab et al. [9] indicate that *P. zhaoi* typically exhibits smaller dimensions (WL 0.66–0.80) in comparison to *P. williamsi* (WL 0.80–0.92). Apart from this size differential, discerning additional morphological distinctions between these species proves challenging.

**Distribution:** CHINA (Yunnan).

**Habitat.** The species was collected in temperate deciduous broad-leaved forest.

## 4. Discussion

As a taxonomic revision synthesizing both historical collections and new material, this study fundamentally reshapes our understanding of *Proceratium* diversity in East Asia. The discovery of seven new species—representing a 54% increase in China’s documented fauna—highlights the remarkable cryptobiotic diversity within this morphologically conservative genus. These findings align with recent studies demonstrating that *Proceratium* species richness has been severely underestimated in subtropical forests [7,9], likely due to their specialized hypogeic lifestyles and microhabitat specificity. The exclusion of *P. longigaster* from China (contra Xu [11]; Staab et al. [9]) resolves a longstanding misidentification, emphasizing the critical importance of type specimen comparisons. The misapplied records instead represent *P. shanyii* and related taxa, underscoring how subtle characters—particularly cephalic sculpture patterns and petiolar node proportions—define species boundaries. The concentration of endemics in Guangxi (4 spp.) and Hainan (2 spp.) mirrors diversity hotspots in other litter-dwelling taxa (e.g., *Strumigenys* Liu et al. [25]). Geographically, the species distributions show some interesting patterns. Several taxa appear restricted to southern China (*P. shanyii*, *P. spinosubum*), while others like *P. japonicum* have wider ranges extending to Japan and Taiwan of China. These patterns may reflect historical biogeographic processes, but more comprehensive sampling is needed to confirm this. Future research should focus on the following: collecting nest series with multiple castes; investigating ecological requirements of different species; applying molecular methods to test species boundaries; and conducting more thorough surveys in poorly studied areas. While this study clarifies some taxonomic issues, much remains to be learned about the biology and evolution of Chinese *Proceratium*. The genus likely holds more surprises for myrmecologists willing to study these cryptobiotic ants in detail.

## Figures and Tables

**Figure 1 insects-16-01060-f001:**
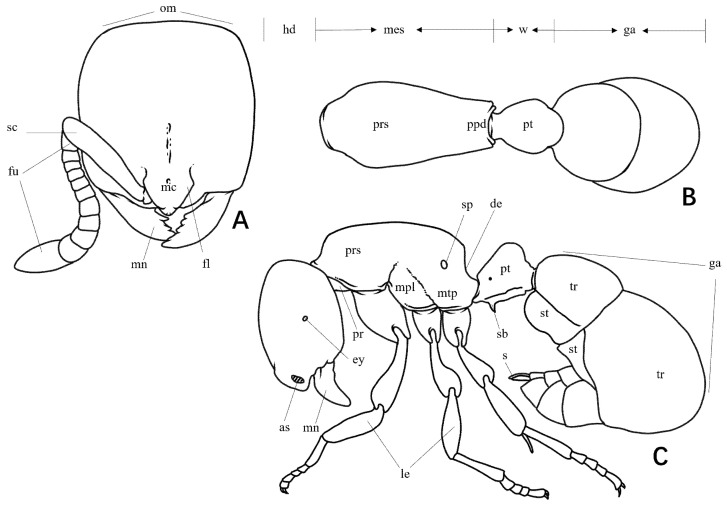
Line drawing of *Proceratium* (as: antennal socket; de: declivity of propodeum; ey: eyes; fl: frontal lobe; fu: funiculi; ga: gaster; hd: head; mes: mesosoma; le: leg; mc: median portion of clypeal; mn: mandible; mpl: mesopleuron; mtp: metapleuron; om: occipital margin; ppd: propodeum; pr: propleuron; prs: promesonotum; pt: petiole; w: waist; s: sting; sb: subpetiolar process; sc: scapes; sp: propodeal spiracle; st: sternite; tr: tergite). (**A**) Head in full-face view; (**B**) body in dorsal view; (**C**) body in lateral view.

**Figure 16 insects-16-01060-f016:**
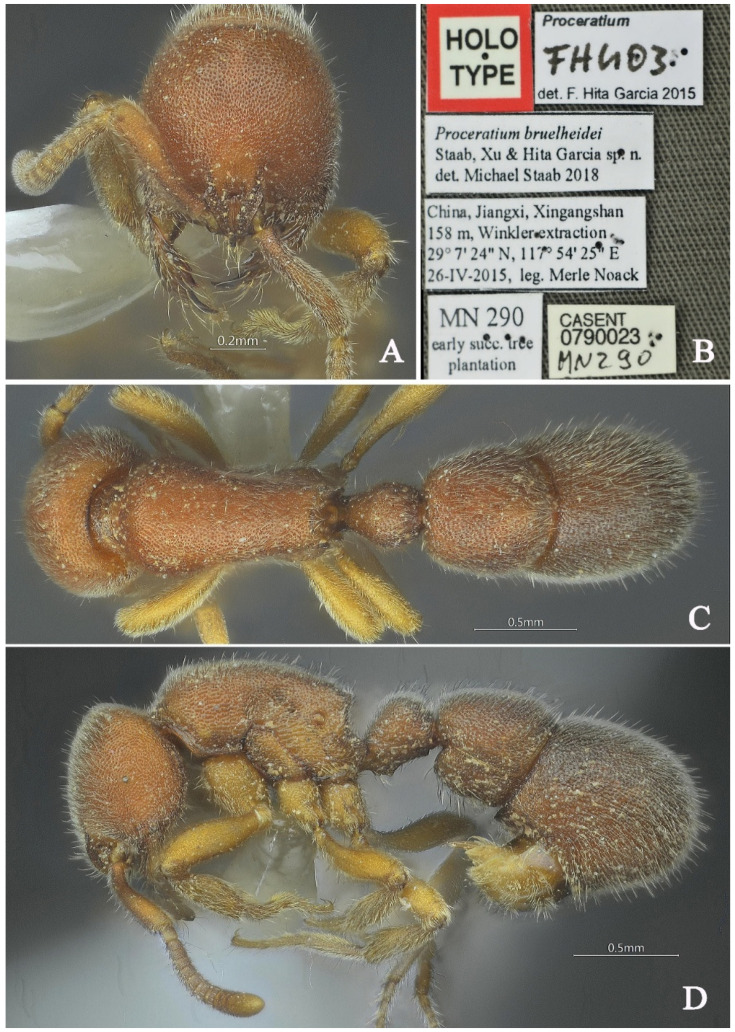
*Proceratium bruelheidei,* holotype worker (photographed by Zhenghui Xu). (**A**) Head in full-face view; (**B**) label of holotype; (**C**) body in dorsal view; (**D**) body in lateral view.

**Figure 17 insects-16-01060-f017:**
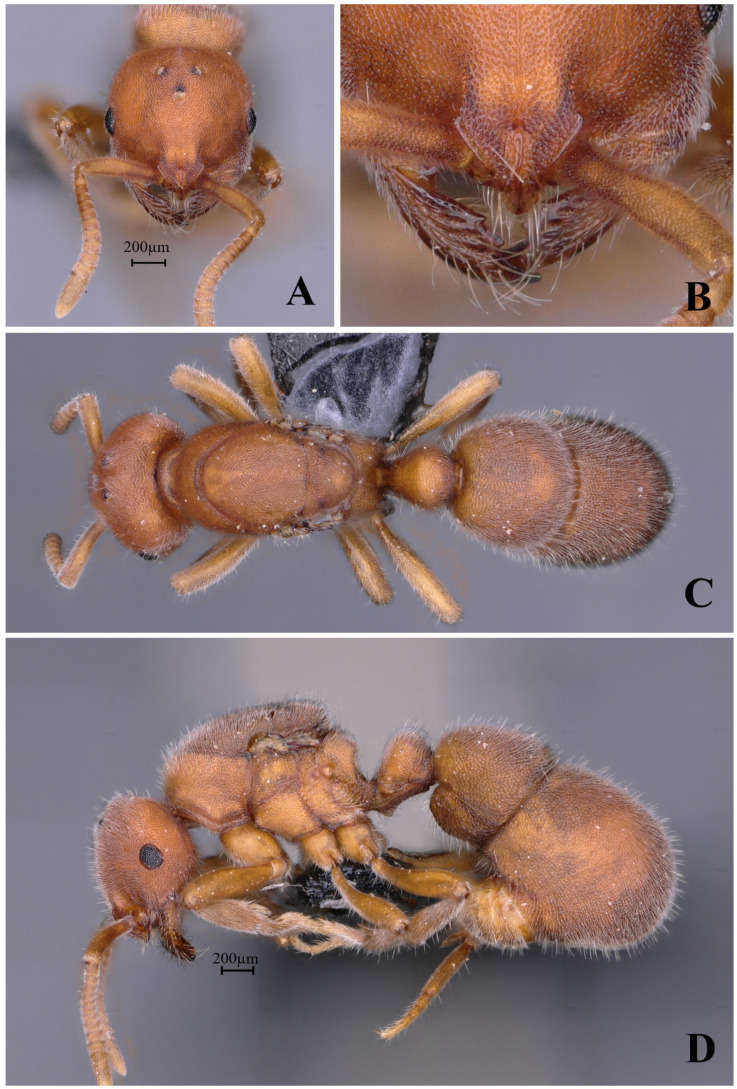
*Proceratium bruelheidei* queen (GXNU110764). (**A**) Head in full-face view; (**B**) mandible in full-face view; (**C**) body in dorsal view; (**D**) body in lateral view.

**Figure 18 insects-16-01060-f018:**
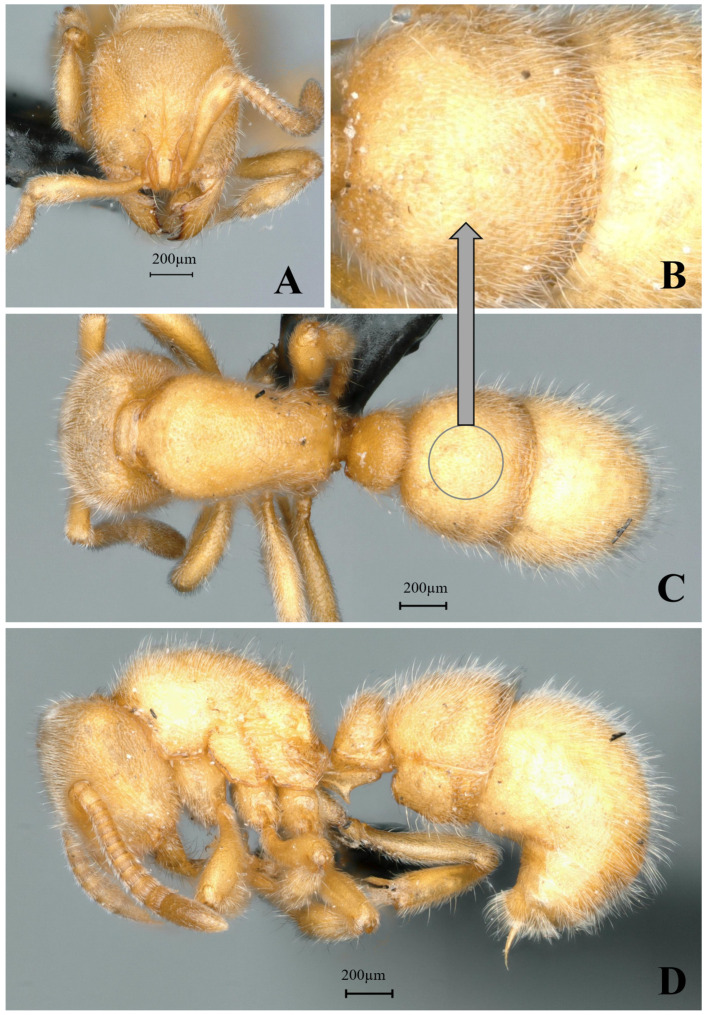
*Proceratium crassopetiolum* sp. nov., holotype worker (photographed by Zhilin Chen). (**A**) Head in full-face view; (**B**) median portion of gastral segment I in dorsal view; (**C**) body in dorsal view; (**D**) body in lateral view (urn:lsid:zoobank.org:act:E018F8FE-8417-4DA7-BC0D-2A305A9E0925).

**Figure 19 insects-16-01060-f019:**
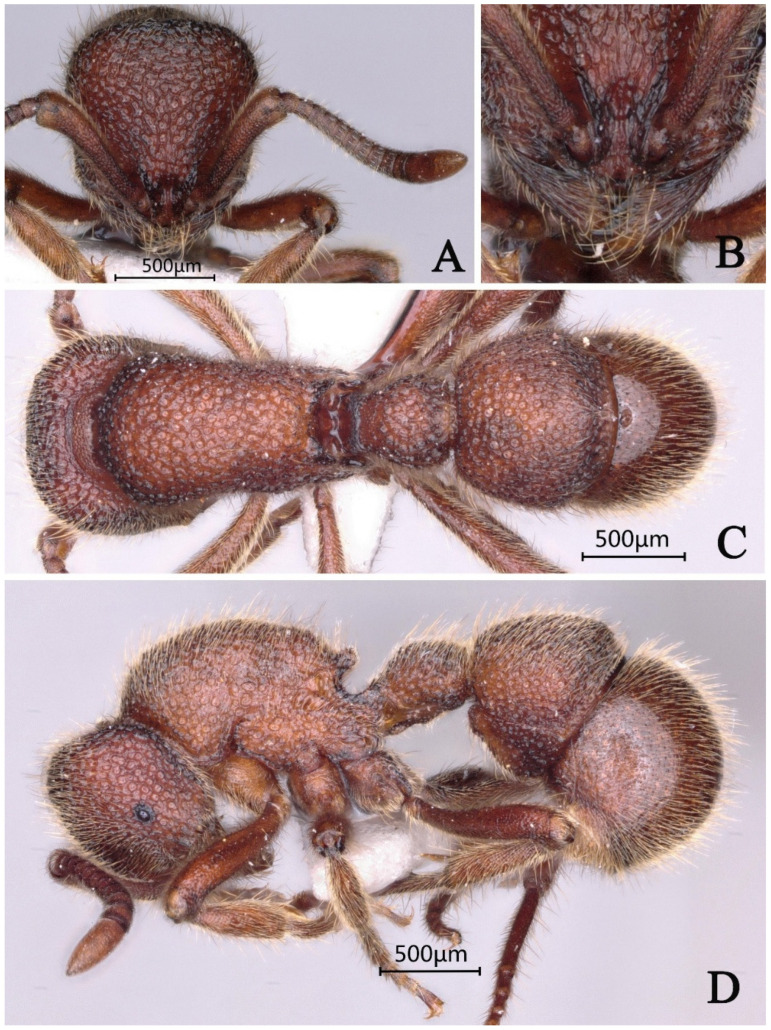
*Proceratium digitospinum* sp. nov., holotype worker (photographed by Zhilin Chen). (**A**) Head in full-face view; (**B**) mandible in full-face view; (**C**) body in dorsal view; (**D**) body in lateral view (urn:lsid:zoobank.org:act:A69D4DF7-C046-4A86-B54C-F7F602A9A76E).

**Figure 20 insects-16-01060-f020:**
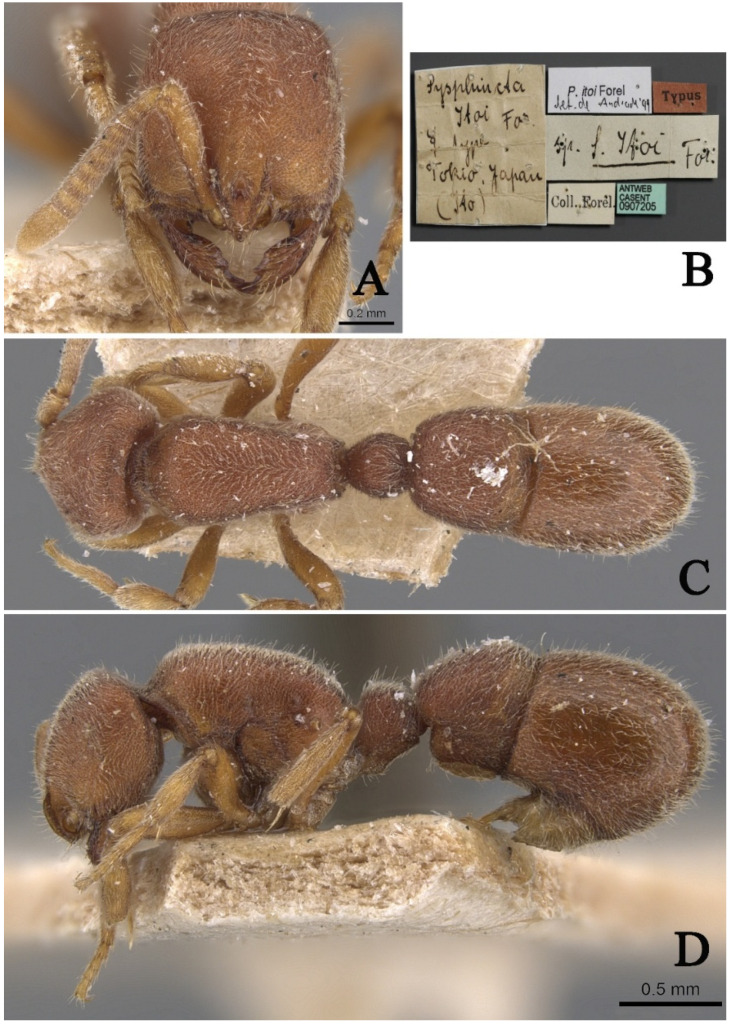
*Proceratium itoi*, syntype worker (https://www.antweb.org, CASENT0907205, photographed by Will Ericson). (**A**) Head in full-face view; (**B**) label of syntype; (**C**) body in dorsal view; (**D**) body in lateral view.

**Figure 21 insects-16-01060-f021:**
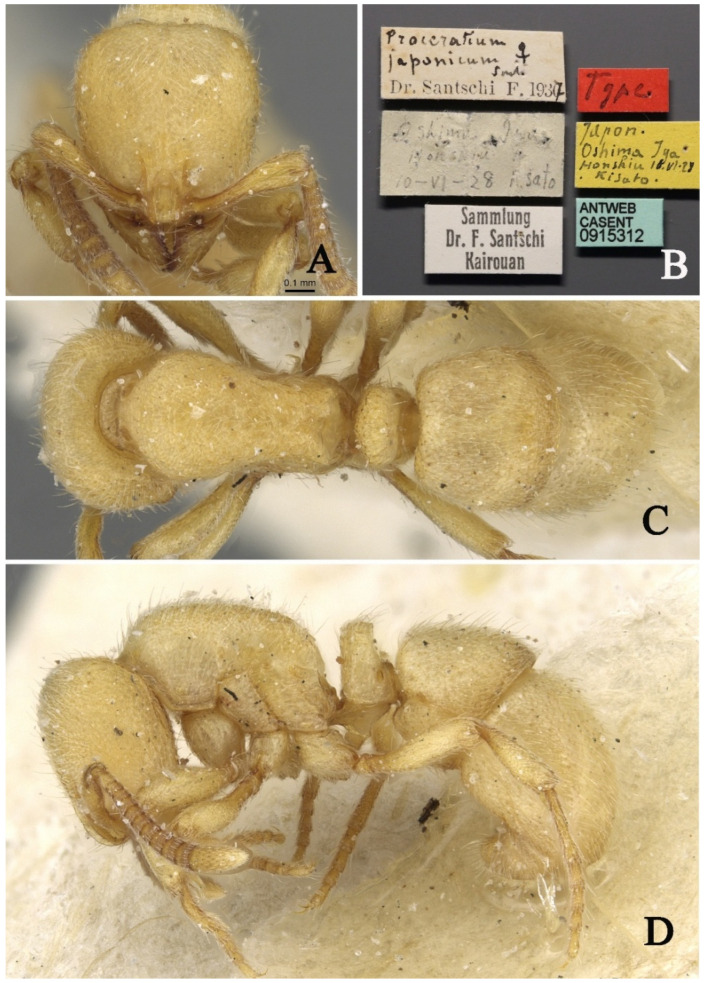
*Proceratium japonicum,* syntype worker (https://www.antweb.org/, CASENT0915312, photographed by Z. Lieberman). (**A**) Head in full-face view; (**B**) label of syntype; (**C**) body in dorsal view; (**D**) body in lateral view.

**Figure 22 insects-16-01060-f022:**
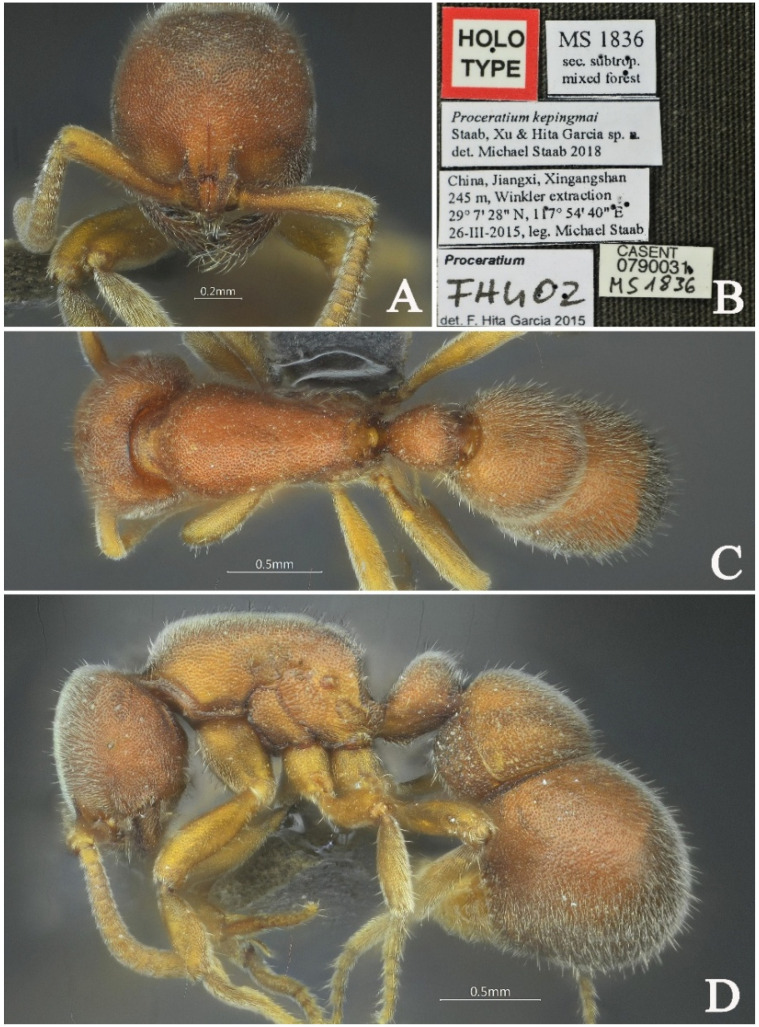
*Proceratium kepingmai,* holotype worker (photographed by Zhenghui Xu). (**A**) Head in full-face view; (**B**) label of holotype; (**C**) body in dorsal view; (**D**) body in lateral view.

**Figure 23 insects-16-01060-f023:**
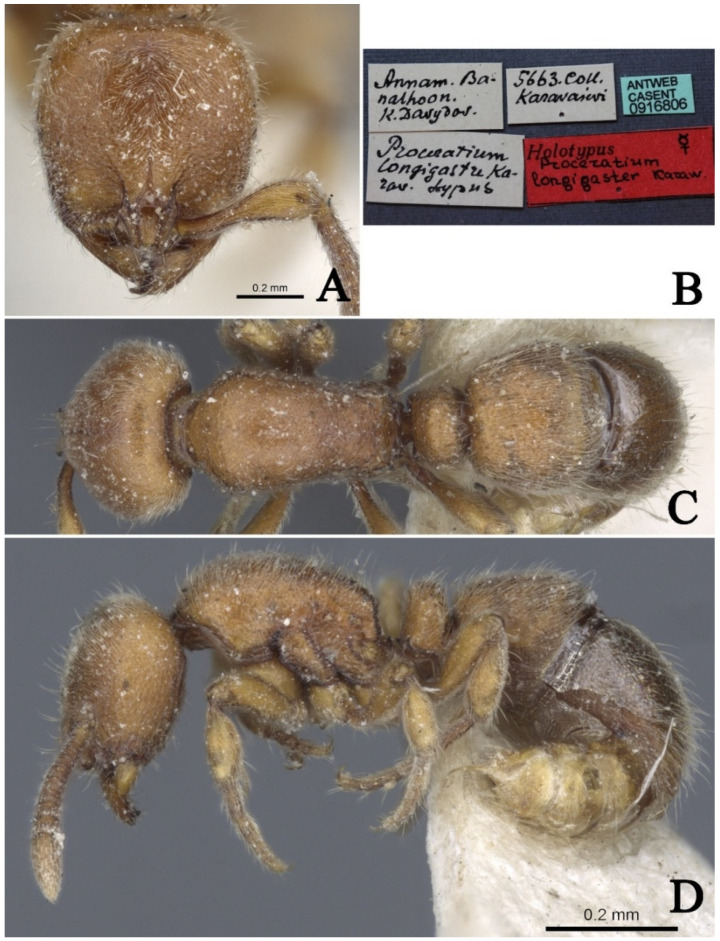
*Proceratium longigaster,* holotype worker (https://www.antweb.org, CASENT0916806, photographed by Flavia Esteves). (**A**) Head in full-face view; (**B**) label of holotype; (**C**) body in dorsal view; (**D**) body in lateral view.

**Figure 24 insects-16-01060-f024:**
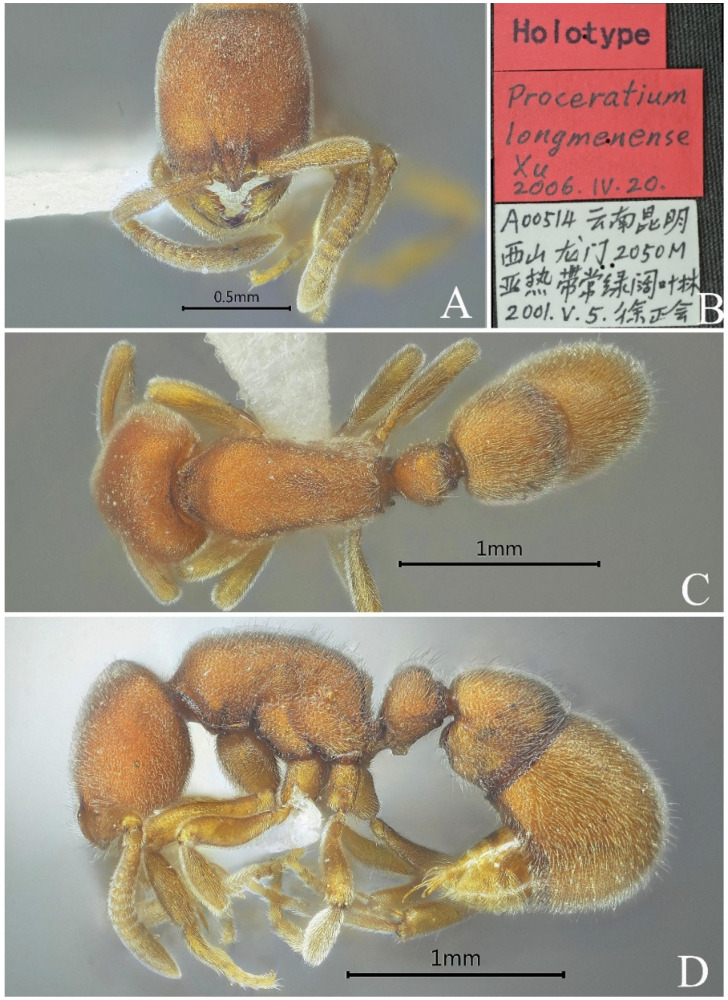
*Proceratium longmenense,* holotype worker (photographed by Zhenghui Xu). (**A**) Head in full-face view; (**B**) label of holotype; (**C**) body in dorsal view; (**D**) body in lateral view.

**Figure 25 insects-16-01060-f025:**
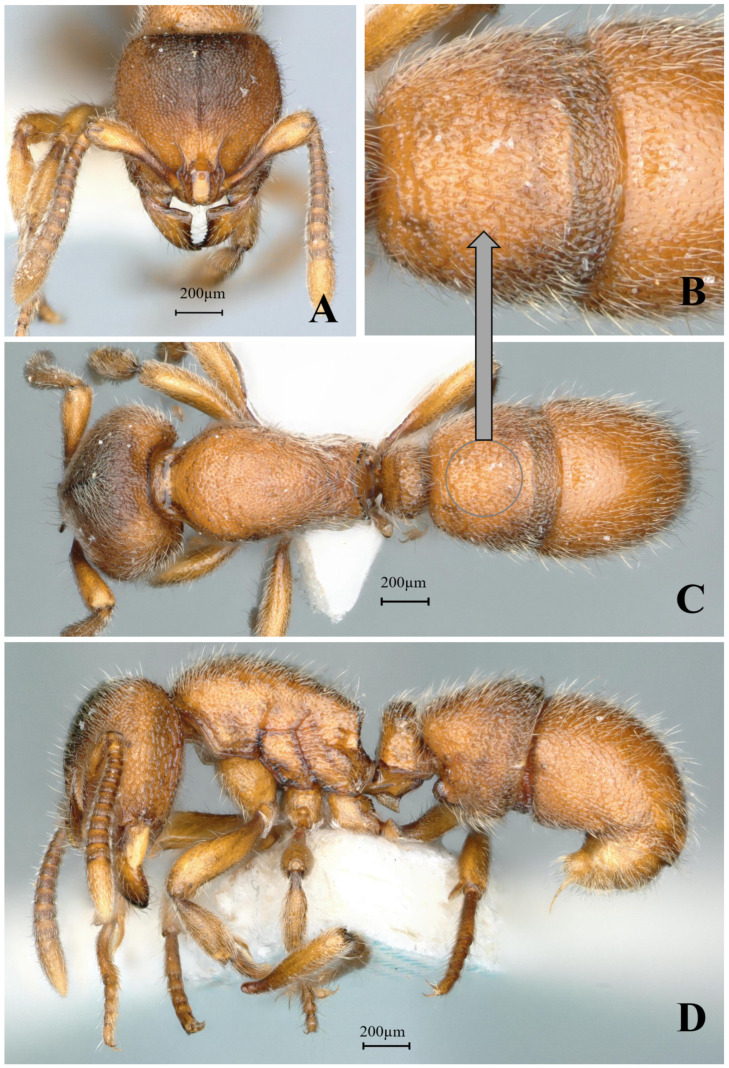
*Proceratium planodorsum* sp. nov., holotype worker (photographed by Zhilin Chen). (**A**) Head in full-face view; (**B**) median portion of gastral segment I in dorsal view; (**C**) body in dorsal view; (**D**) body in lateral view (urn:lsid:zoobank.org:act:06512620-57ED-4B17-B5D3-B266BD75477F).

**Figure 26 insects-16-01060-f026:**
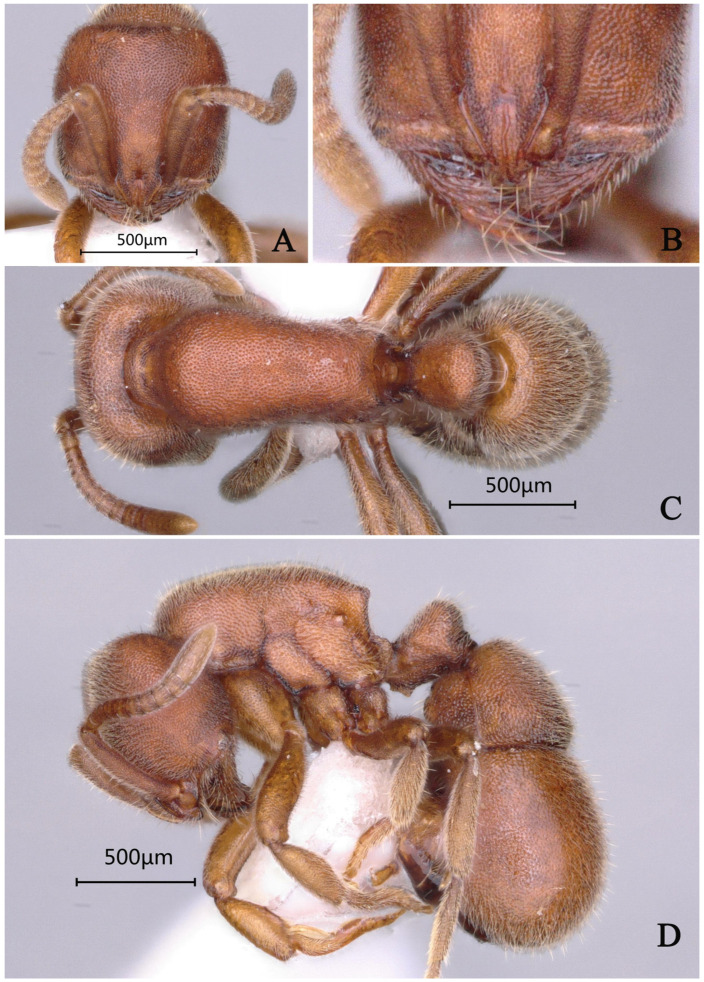
*Proceratium recticephalum* sp. nov., holotype worker (photographed by Zhilin Chen). (**A**) Head in full-face view; (**B**) mandible in full-face view; (**C**) body in dorsal view; (**D**) body in lateral view (urn:lsid:zoobank.org:act:F4BA392D-35BE-4D36-856D-316B91FFE6B7).

**Figure 27 insects-16-01060-f027:**
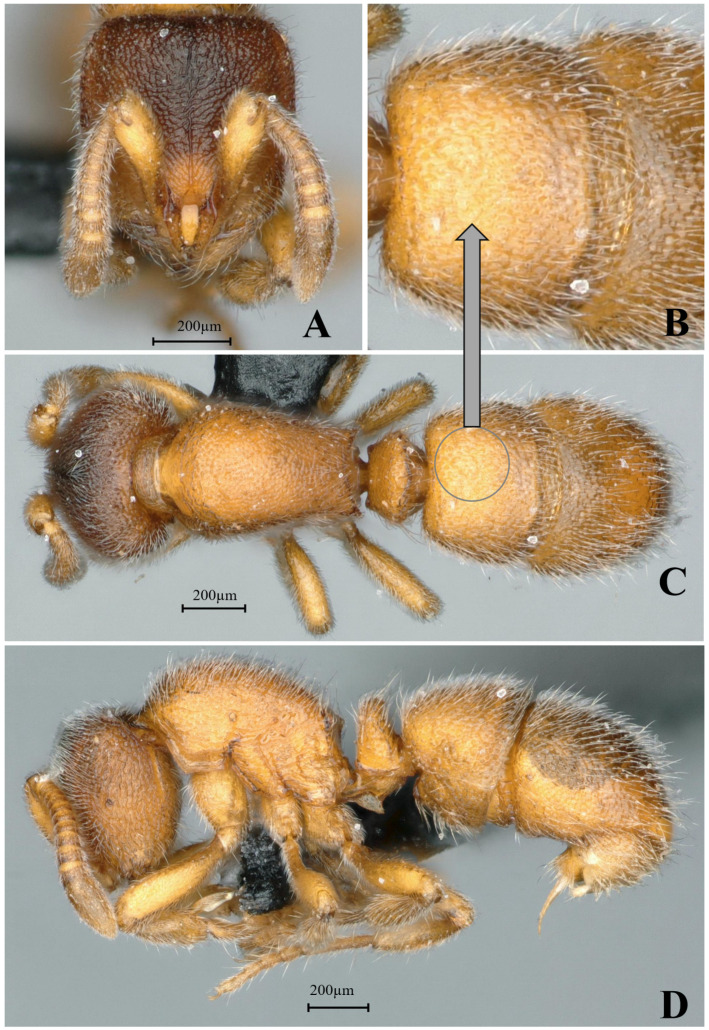
*Proceratium rugiceps* sp. nov., holotype worker (photographed by Zhilin Chen). (**A**) Head in full-face view; (**B**) median portion of gastral segment I in dorsal view; (**C**) body in dorsal view; (**D**) body in lateral view (urn:lsid:zoobank.org:act:E1E02369-3382-4072-A810-D8B9CC1C06F6).

**Figure 28 insects-16-01060-f028:**
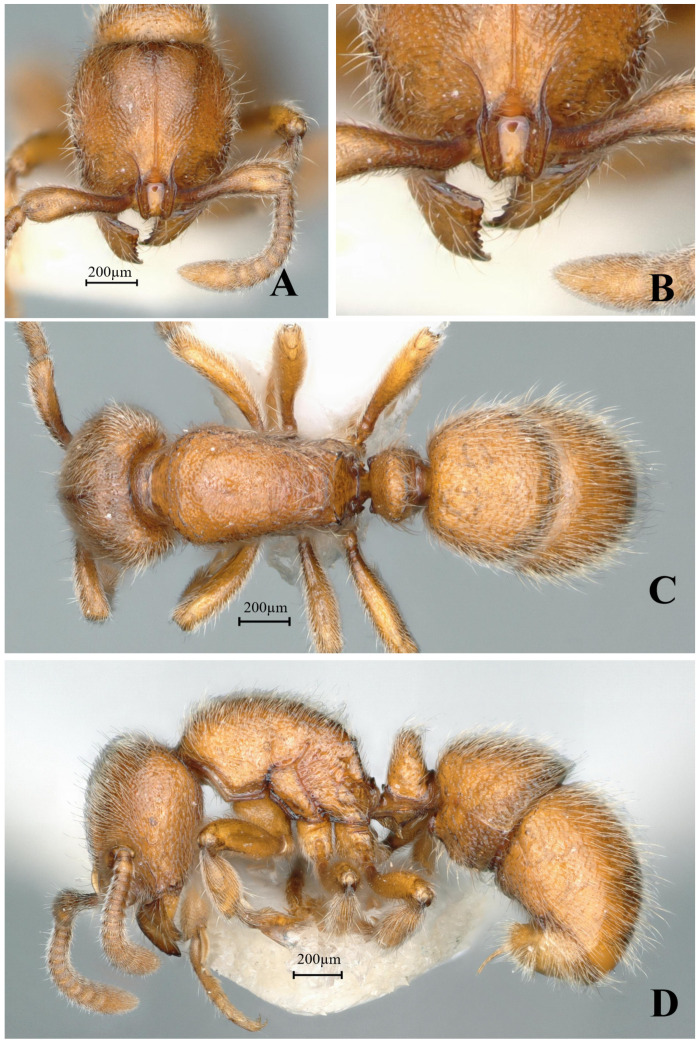
*P. shanyii* sp. nov., holotype worker (photographed by Zhilin Chen). (**A**) Head in full-face view; (**B**) mandible in anterodorsal view; (**C**) body in dorsal view; (**D**) body in lateral view (urn:lsid:zoobank.org:act:EDB37856-7E09-4EBA-8F0A-A4AA9EEA48DE).

**Figure 29 insects-16-01060-f029:**
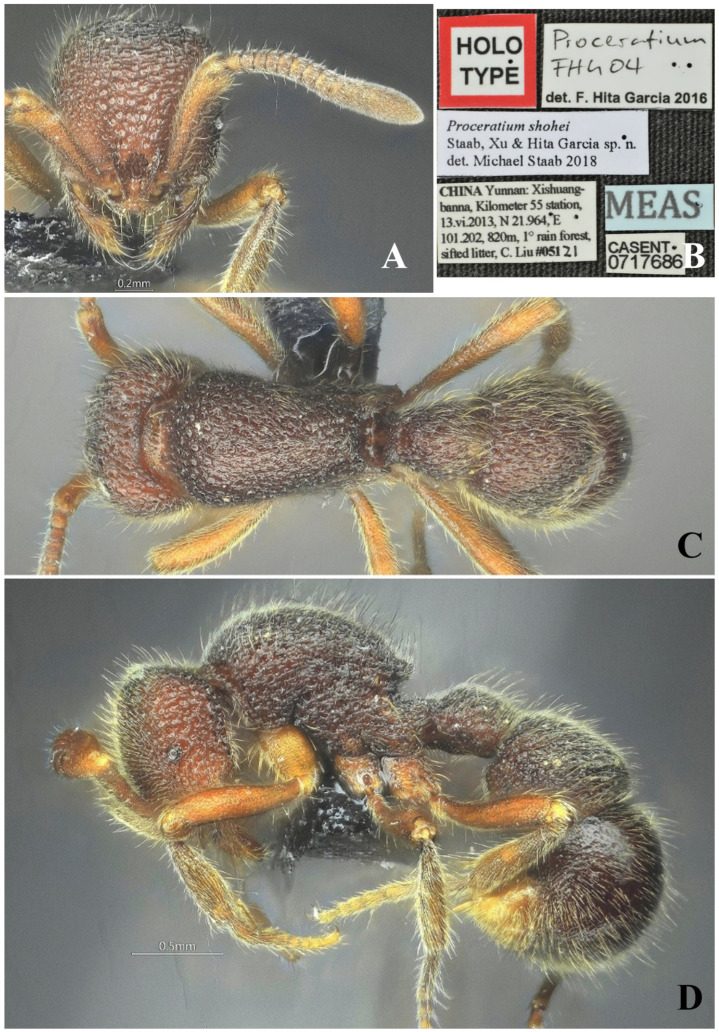
*Proceratium shohei*, holotype worker (photographed by Zhenghui Xu). (**A**) Head in full-face view; (**B**) label of holotype; (**C**) body in dorsal view; (**D**) body in lateral view.

**Figure 30 insects-16-01060-f030:**
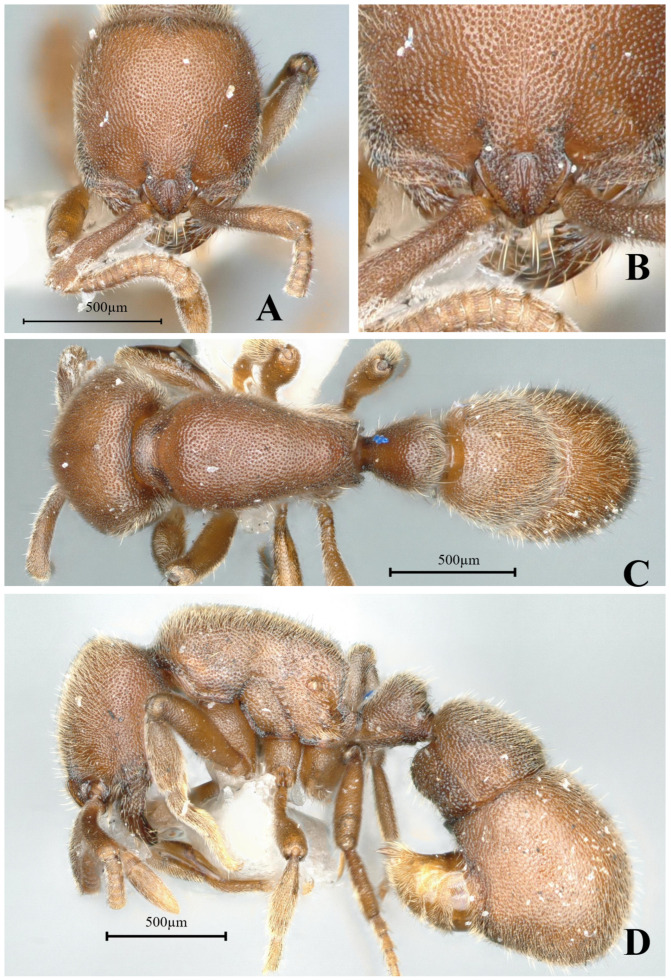
*Proceratium spinosubum* sp. nov., holotype worker (photographed by Zhilin Chen). (**A**) Head in full-face view; (**B**) frontal carinae in full-face view; (**C**) body in dorsal view; (**D**) body in lateral view (urn:lsid:zoobank.org:act:3C22F275-8527-4BA3-BDB1-C37E5A315507).

**Figure 31 insects-16-01060-f031:**
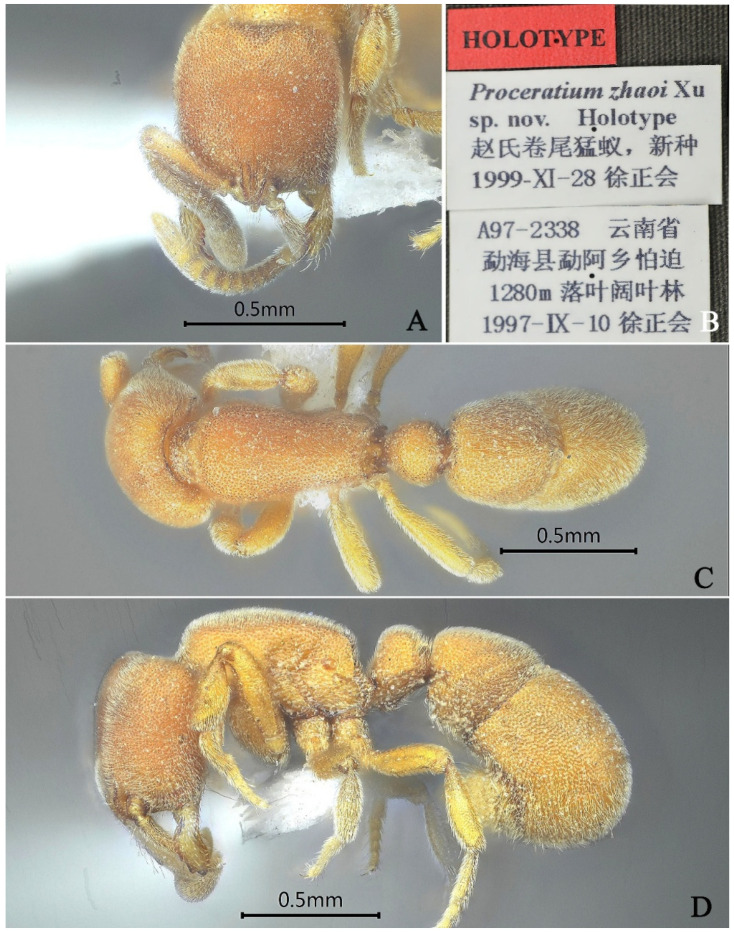
*Proceratium zhaoi,* holotype worker. (**A**) Head in full-face view; (**B**) label of holotype; (**C**) body in dorsal view; (**D**) body in lateral view.

## Data Availability

Specimens are available at the Guangxi Normal University, China and Southwest Forestry University, China; the examined images of three species (*P. itoi*, *P. japonicum, P. longigaster*) are cited from AntWeb (https://www.antweb.org).

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
