# Peer review of "Synopsis of Ant Genus Proceratium Roger, 1863 from China (Hymenoptera, Formicidae), with Description of Seven New Speciesâ€"

_insects, 2025, doi:10.3390/insects16101060_

Round 1
Reviewer 1 Report
Comments and Suggestions for Authors
I strongly advise registering the manuscript and new species in Zoobank and including the code in the manuscript.
And please consider the term "cryptic species/genus" once again. Because I know you want to mention the hiding species with this term. But this term refers to a different meaning. I indicate it in the manuscript.
According to my search queen of Proceratiom bruelheidei has not yet been described. You found 1 queen in Tianmu Shan, Zejian. You should describe the queen for the first time and include it in the manuscript. This inclusion will raise the quality of the manuscript.
My other suggestions are in the PDF file.

Author Response
We sincerely appreciate your thoughtful review and constructive feedback on our manuscript entitled “Synopsis of ant genus Proceratium Roger, 1863 from China (Hymenoptera, Formicidae), with description of seven new species". Your insights have been instrumental in refining our work and improving its clarity and scientific rigor. Below, we address each of your comments and suggestions in detail:
Reviewer Comment 1:
Comment: “Because of taxonomical act occures in the text, manuscript should be registered to zoobank, and should be indicated the zoobank code.”
“I strongly advise registering the manuscript and new species in Zoobank and including the code in the manuscript.”
Response: See rows 109–111. Thank you very much for your insightful comments. We appreciate your recognition of the scientific contributions of our paper and your suggestions for its improvement. We have registered all the seven new species to zoobank and updated the zoobank code in the manuscript: “This article is registered in ZooBank (urn:lsid:zoobank.org:pub:E3249182-A18A-4A78-8AFD-289499DA0B29) in compli-ance with ICZN requirements for electronic publications. The ZooBank registration was completed prior to final publication.”.
Reviewer Comment 2:
Comment: “P. spinosubum should be add here.”
Response: See rows 29. We sincerely apologize for the mistake. Thank you for pointing it out. We have supplemented P. spinosubum.
Reviewer Comment 3:
Comment: “Cryptic species/genus meaning morphologically very similar or indistinguishable, but can be dfferentiated by genetic, ecological, or behavioral traits. Therefore I suggest to use "cryptobiotic genus" here.”
“And please consider the term "cryptic species/genus" once again. Because I know you want to mention the hiding species with this term. But this term refers to a different meaning. I indicate it in the manuscript.”
Response: See rows 32. Thank you for your insightful comments. We have updated the description to accurately state "cryptobiotic" instead of "cryptic".
Reviewer Comment 4:
Comment: “Abstract should be started with a short background information.”
Response: See rows 34–39. Thank you for your comment regarding abstract. We have expanded the background information: “the genus Proceratium comprises rare but ecologically significant “cryptobiotic” pred-ators of temperate and tropical forest litter. Named for an abdomen that can curl dorsally >90° relative to the body axis, the group includes 130 described species worldwide. In China, seven species have been recorded, yet recent surveys repeatedly reveal mor-phologically distinctive undescribed taxa, indicating a still-underestimated diversity of Proceratium in the country.”
Reviewer Comment 5:
Comment: “Unnecessary information.”
Response: See rows 40–42. Thank you for pointing this out. Following your suggestions, we have removed the unnecessary information.
Reviewer Comment 6:
Comment: “87.”
Response: See rows 50. We sincerely apologize for the mistake. We have corrected it. Thank you for pointing it out.
Reviewer Comment 7:
Comment: “Author can be refer only Bolton Catalogue in AntCat (see https://antcat.org/catalog/430157?qq=Proceratium).”
Response: See rows 51, 52. Thank you for your feedback. We have checked the list of authors in Antcat, and we have corrected the reference.
Reviewer Comment 8:
Comment: “This information can be found with a simple literature search. Therefore no need to give a citation.”
Response: See rows 59. Thank you for your suggestion. We have removed the unnecessary citation.
Reviewer Comment 9:
Comment: “Must be uperscript.”
Response: See rows 59. Thank you for your comment. We sincerely apologize for making the mistake. We have corrected it.
Reviewer Comment 10:
Comment: “Author should be stated where the other examined specimens come from.”
Response: See rows 73–75. Thank you for your suggestion. We have updated and supplemented the information of the examined specimens.
Reviewer Comment 11:
Comment: “Seven.”
Response: See rows 76. Thank you for pointing it out. We have corrected it.
Reviewer Comment 12:
Comment: “I recommended to add the name of the measurements and indexes like: Cephalic Indes (Cl) = HW x 1oo / HL. And Measurements and indexes should be seperate under small subheaders.”
Response: See rows 81–108. Thank you very much for your insightful comments. We have updated the clear definition about those measurements and indexes.
Reviewer Comment 13:
Comment: “Delete it.”
Response: See rows 83. Thank you for your comments. We have made the revisions as you requested to remove this literature.
Reviewer Comment 14:
Comment: “Need reference.”
Response: See rows 123. Thank you for your valuable comments. We have updated the reference here.
Reviewer Comment 15:
Comment: “Please do not shorten the genus name when it located at the begining of a sentence.”
Response: See rows 142. Thank you for your comments. We have supplemented the genus name.
Reviewer Comment 16:
Comment: “Of mandibles.”
Response: See rows 175. Thank you for your suggestions. We have made the revisions as you requested to update it.
Reviewer Comment 17:
Comment: “Of mandibles.”
Response: See rows 181. Thank you for your suggestions. We have made the revisions as you requested to update it.
Reviewer Comment 18:
Comment: “Long.”
Response: See rows 214. Thank you for your feedback. We have corrected it.
Reviewer Comment 19:
Comment: “Long.”
Response: See rows 215. Thank you for your feedback. We have corrected it.
Reviewer Comment 20:
Comment: “(The hairs and pubescens on the scape have been intentionally reduced).”
Response: See rows 218–225. Thank you very much for your insightful comments. We have removed this unclear character.
Reviewer Comment 21:
Comment: “Leg.”
Response: See rows 264. Thank you for your comments. We have made the revisions as you requested to updated it.
Reviewer Comment 22:
Comment: “The queen of P. bruelheidei is not determined yet. Author should describe the queen detaily here too.”
Response: See rows 277–301, rows 307–309. Thank you very much for your insightful comments. We have updated the information and images of the P. bruelheidei queen.
Reviewer Comment 23:
Comment: “Not 680m. It high is as around 360m. Check again.”
Response: See rows 265. Thank you very much for your comments. We have checked the label information and corrected it.
Reviewer Comment 24:
Comment: “Leg.”
Response: See rows 265. Thank you for your comments. We have made the revisions as you requested to updated it.
Reviewer Comment 25:
Comment: “Please do not shorten the genus name when it located at the begining of a sentence.”
Response: See rows 304. Thank you very much for your insightful comments. We have corrected it.
Reviewer Comment 26:
Comment: “Please register the species to zoobank and provide here a code.”
Response: See rows 376–377. Thank you very much for your insightful comments. We have registered the new species to zoobank and updated the zoobank code in the manuscript: “urn:lsid:zoobank.org:act:E018F8FE-8417-4DA7-BC0D-2A305A9E0925”.
Reviewer Comment 27:
Comment: “(n=7).”
Response: See rows 315. Thank you very much for your insightful comments. We have corrected it.
Reviewer Comment 28:
Comment: “There is no scale.”
Response: See rows 373. Thank you very much for pointing it out. We have made the images as you requested to add the scales.
Reviewer Comment 29:
Comment: “Please register the species to zoobank and provide here a code.”
Response: See rows 430. Thank you very much for your insightful comments. We have registered the new species to zoobank and updated the zoobank code in the manuscript: “urn:lsid:zoobank.org:act:A69D4DF7-C046-4A86-B54C-F7F602A9A76E”.
Reviewer Comment 30:
Comment: “Around 160m. Check again”
Response: See rows 381. Thank you very much for your insightful comments. We have checked and corrected it.
Reviewer Comment 31:
Comment: “Leg.”
Response: See rows 381. Thank you for your comments. We have made the revisions as you requested to updated it.
Reviewer Comment 32:
Comment: “Leg.”
Response: See rows 486. Thank you for your comments. We have made the revisions as you requested to updated it.
Reviewer Comment 33:
Comment: “Vietnam.”
Response: See rows 535. Thank you very much for pointing it out. We have made the revisions as you requested.
Reviewer Comment 34:
Comment: “Leg.”
Response: See rows 546. Thank you for your comments. We have made the revisions as you requested to updated it.
Reviewer Comment 35:
Comment: “Please register the species to zoobank and provide here a code.”
Response: See rows 623–624. Thank you very much for your insightful comments. We have registered the new species to zoobank and updated the zoobank code in the manuscript: “urn:lsid:zoobank.org:act:06512620-57ED-4B17-B5D3-B266BD75477F”.
Reviewer Comment 36:
Comment: “Around 230m. Check again”
Response: See rows 565. Thank you for your comments. We have checked the label information and corrected it.
Reviewer Comment 37:
Comment: “Leg.”
Response: See rows 561. Thank you for your comments. We have made the revisions as you requested to updated it.
Reviewer Comment 38:
Comment: “There is no scale.”
Response: See rows 620. Thank you very much for pointing it out. We have made the images as you requested to add the scale.
Reviewer Comment 39:
Comment: “Please register the species to zoobank and provide here a code.”
Response: See rows 690. Thank you very much for your insightful comments. We have registered the new species to zoobank and updated the zoobank code in the manuscript: “urn:lsid:zoobank.org:act:F4BA392D-35BE-4D36-856D-316B91FFE6B7”.
Reviewer Comment 40:
Comment: “~200m.”
Response: See rows 628. Thank you for your comments. We have checked the label information and corrected it.
Reviewer Comment 41:
Comment: “Leg.”
Response: See rows 628. Thank you for your comments. We have made the revisions as you requested to updated it.
Reviewer Comment 42:
Comment: “Please register the species to zoobank and provide here a code.”
Response: See rows 745. Thank you very much for your insightful comments. We have registered the new species to zoobank and updated the zoobank code in the manuscript: “urn:lsid:zoobank.org:act:E1E02369-3382-4072-A810-D8B9CC1C06F6”.
Reviewer Comment 43:
Comment: “~920m.”
Response: See rows 694. Thank you for your comments. We have checked the label information and corrected it.
Reviewer Comment 44:
Comment: “Leg.”
Response: See rows 694. Thank you for your comments. We have made the revisions as you requested to updated it.
Reviewer Comment 45:
Comment: “Please register the species to zoobank and provide here a code.”
Response: See rows 800. Thank you very much for your insightful comments. We have registered the new species to zoobank and updated the zoobank code in the manuscript: “urn:lsid:zoobank.org:act:EDB37856-7E09-4EBA-8F0A-A4AA9EEA48DE”.
Reviewer Comment 46:
Comment: “~160m.”
Response: See rows 750. Thank you for your comments. We have checked the label information and corrected it.
Reviewer Comment 47:
Comment: “Leg.”
Response: See rows 750. Thank you for your comments. We have made the revisions as you requested to updated it.
Reviewer Comment 48:
Comment: “Scales are forgotten in the plates.”
Response: See rows 797. Thank you very much for pointing it out. We have updated the images as you requested to add the scales.
Reviewer Comment 49:
Comment: “Please register the species to zoobank and provide here a code.”
Response: See rows 877. Thank you very much for your insightful comments. We have registered the new species to zoobank and updated the zoobank code in the manuscript: “urn:lsid:zoobank.org:act:3C22F275-8527-4BA3-BDB1-C37E5A315507”.
Reviewer Comment 50:
Comment: “Leg.”
Response: See rows 833. Thank you for your comments. We have made the revisions as you requested to updated it.
Reviewer Comment 51:
Comment: “Scale?”
Response: See rows 874. Thank you very much for pointing it out. We have made the revisions as you requested to change the images.
Reviewer Comment 52:
Comment: “–.”
Response: See rows 898. Thank you very much. We sincerely apologize for making the mistake. We have corrected and supplemented it.
Reviewer Comment 53:
Comment: “I suggested to use here "Cryptobiotic"”
Response: See rows 898. Thank you for your insightful comments. We have updated the description to accurately state "cryptobiotic" instead of "cryptic".
Reviewer Comment 54:
Comment: “Cryptobiotic.”
Response: See rows 917. Thank you for your insightful comments. We have updated the description to accurately state "cryptobiotic" instead of "cryptic".
Reviewer 2 Report
Comments and Suggestions for Authors
Comments and questions to the authors:
Material and methods
I suggest to give the total number of the specimens of the new species in the first paragraph.
Results
- Row 389-391 – the “locus typicus”, geographical coordinates and date of assessment of AntWeb is good to be added for the readers’ convenience.
- The same information is lacking also for other species in rows: 420-422, 453-455, 742-743.
- Notes about the biology of the new described species would be of great ecological importance. As hypogeic, cryptic species, comprehensive information is hard to be given but even a little information is of great importance. At least the authors may give the habitats for Proceratium shanyii nov., Proceratium rugiceps sp. nov., Proceratium spinosubum sp. nov. as well as for few of the already published species where the habitat data is lacking.
- Molecular sequences of the new described species would be of great importance for the myrmecological science. Have you taken any steps towards this goal?
References
- Row 837: the cited paper Liu et al. (2020) is not in References.
- Rows 883-886: there are two articles of Liu et al. 2015. They have to be separated by the letters “2015a” and “2015b” in the References and also in the text.
- Row 891: Perrault G.H. 1981 is not cited in the text of the manuscript.
- Rows: 907-908: the paper of Tang et al. 1995 has to be deleted as it is given on the rows 898-899.
Author Response
We sincerely appreciate your thoughtful review and constructive feedback on our manuscript entitled “Synopsis of ant genus Proceratium Roger, 1863 from China (Hymenoptera, Formicidae), with description of seven new species”. Your insights have been instrumental in refining our work and improving its clarity and scientific rigor. Below, we address each of your comments and suggestions in detail:
Reviewer Comment 1:
Comment: “I suggest to give the total number of the specimens of the new species in the first paragraph.”
Response: See rows 77. Thank you very much for your insightful comments. We have add the stance “A total of 28 specimens of the seven new species were collected” in the materials and methods.
Reviewer Comment 2:
Comment: “Row 389–391 – the “locus typicus”, geographical coordinates and date of assessment of AntWeb is good to be added for the readers’ convenience.”
Response: See rows 436–437. Thank you very much for your comments. We have made the revisions as you requested to supplement the “locus typicus”, geographical coordinates and date of assessment of Antweb.
Reviewer Comment 3:
Comment: “The same information is lacking also for other species in rows: 420-422, 453–455, 742–743.”
Response: See rows 471–472, rows 507–508, rows 806–807. Thank you for your comment. Following your suggestions, we have supplemented the information of other species.
Reviewer Comment 4:
Comment: “Notes about the biology of the new described species would be of great ecological importance. As hypogeic, cryptic species, comprehensive information is hard to be given but even a little information is of great importance. At least the authors may give the habitats for Proceratium shanyii sp. nov., Proceratium rugiceps sp. nov., Proceratium spinosubum sp. nov. as well as for few of the already published species where the habitat data is lacking.”
Response: See rows 300, rows 458–459, rows 477, rows 495, rows 536–537, rows 554, rows 736–737, rows 793, rows 824–825, rows 869, rows 890. Thank you for your professional comments. Following your suggestions, we have updated the information of habitats about P. shanyii sp. nov., P. rugiceps sp. nov., P. spinosubum sp. nov and the known species.
Reviewer Comment 5:
Comment: “Molecular sequences of the new described species would be of great importance for the myrmecological science. Have you taken any steps towards this goal?”
Response: We sincerely thank the reviewer for this crucial comment. We fully agree and acknowledge that integrating molecular data is a cornerstone of modern taxonomic practice, and its absence is a significant limitation of our current study, for which we apologize.
The primary reason in which the specimens were preserved and their limited availability. Regrettably, the type series of this new species is extremely limited. To safeguard the integrity of the voucher material on which the description is based-and to ensure its future availability for morphological examination-all type specimens were prepared as pinned and dried material following standard protocols for myrmecological collections. Only in hindsight did we realize that this method is incompatible with downstream DNA extraction and preservation, particularly for minute organisms, as DNA degradation in dried specimens is a well-documented and formidable challenge. Nevertheless, we have not dismissed the reviewer’s concern.
Our most concrete commitment is to conduct a dedicated field expedition to the type locality specifically aimed at collecting fresh material that will be immediately preserved in absolute ethanol to ensure optimal DNA preservation. Should we succeed in recollecting the species, generating molecular sequences will be our top priority, and these sequences will be deposited in open-access repositories such as GenBank for the benefit of the scientific community.
We respectfully hope the reviewer to understand these extenuating circumstances. While we deeply regret the current lack of molecular data, we trust that the comprehensive morphological evidence presented herein provides a solid foundation for the recognition of this new taxon. We are very grateful for the reviewer’s feedback, which has directly guided our ongoing and future research plans for this species.
Reviewer Comment 6:
Comment: “Row 837: the cited paper Liu et al. (2020) is not in References.”
Response: See rows 958–960. We sincerely apologize for the mistake. We have corrected it. Thank you for pointing it out. We have updated the references.
Reviewer Comment 7:
Comment: “Rows 883–886: there are two articles of Liu et al. 2015. They have to be separated by the letters “2015a” and “2015b” in the References and also in the text.”
Response: See Rows 954–957. Thank you for your suggestion regarding the reference. We have updated the letters “2015a” and “2015b” to separate the two citations in the manuscript.
Reviewer Comment 8:
Comment: “Row 891: Perrault G.H. 1981 is not cited in the text of the manuscript.”
Response: See Rows 61. Thank you for your comment. We have rechecked the list of reference with certification to avoid making the mistake, and we have updated it.
Reviewer Comment 9:
Comment: “Rows: 907-908: the paper of Tang et al. 1995 has to be deleted as it is given on the rows 898-899.”
Response: See rows 985. We sincerely apologize for the mistake. We have removed the redundant reference. Thank you very much for pointing it out.
Round 2
Reviewer 1 Report
Comments and Suggestions for Authors
The manuscript is written well. The photos are high-quality and informative. The descriptions are taxonomically appropriate. Congratulations, you added quite a huge new Proceratium species for science and the Chinese ant fauna.
Author Response
We have submitted our revised manuscript (Manuscript ID: insects-3845919). However, we have not received any further comments or instructions from the reviewers for this second round of review. The field for the reviewers' comments appears to be blank in the submission system.
We are ready to address any additional feedback should it be required.
Thank you for your attention to this matter.
Sincerely,
Zhilin Chen